# *Salmonella* Typhimurium reprograms macrophage metabolism via T3SS effector SopE2 to promote intracellular replication and virulence

Lingyan Jiang [1,2,6], Peisheng Wang[1,2,6], Xiaorui Song[1,2,6], Huan Zhang[1,2], Shuangshuang Ma[1,2], Jingting Wang[1,2], Wanwu Li[1,2], Runxia Lv[1,2], Xiaoqian Liu[1,2], Shuai Ma[1,2], Jiaqi Yan[3], Haiyan Zhou[4], Di Huang[1,2], Zhihui Cheng[1,3], Chen Yang[4], Lu Feng [1,2✉] & Lei Wang [1,2,5✉]

*Salmonella* Typhimurium establishes systemic infection by replicating in host macrophages. Here we show that macrophages infected with *S.* Typhimurium exhibit upregulated glycolysis and decreased serine synthesis, leading to accumulation of glycolytic intermediates. The effects on serine synthesis are mediated by bacterial protein SopE2, a type III secretion system (T3SS) effector encoded in pathogenicity island SPI-1. The changes in host metabolism promote intracellular replication of *S.* Typhimurium via two mechanisms: decreased glucose levels lead to upregulated bacterial uptake of 2- and 3-phosphoglycerate and phosphoenolpyruvate (carbon sources), while increased pyruvate and lactate levels induce upregulation of another pathogenicity island, SPI-2, known to encode virulence factors. Pharmacological or genetic inhibition of host glycolysis, activation of host serine synthesis, or deletion of either the bacterial transport or signal sensor systems for those host glycolytic intermediates impairs *S.* Typhimurium replication or virulence.

[1] The Key Laboratory of Molecular Microbiology and Technology, Ministry of Education, Nankai University, Tianjin, China. [2] TEDA Institute of Biological Sciences and Biotechnology, Tianjin Key Laboratory of Microbial Functional Genomics, Nankai University, Tianjin, China. [3] College of Life Sciences, Nankai University, Tianjin, China. [4] CAS-Key Laboratory of Synthetic Biology, CAS Center for Excellence in Molecular Plant Sciences, Shanghai Institute of Plant Physiology and Ecology, Chinese Academy of Sciences, Shanghai, China. [5] The Institute of Translational Medicine Research, Tianjin Union Medical Center, Nankai University Affiliated Hospital, Nankai University, Tianjin, China. [6] These authors contributed equally: Lingyan Jiang, Peisheng Wang, Xiaorui Song. ✉email: fenglu63@nankai.edu.cn; wanglei@nankai.edu.cn

  1

Salmonella enterica serovar Typhimurium (STM) is an important intracellular pathogen that causes gastroenteritis and life-threatening systemic disease in humans and animals[1]. It invades multiple cell types during infection, including dendritic cells, epithelial cells, and macrophages. Although macrophages generally represent the front-line host defense against invading bacterial pathogens, they are a crucial colonization niche during STM pathogenesis. The replication in macrophages enables STM to establish systemic disease in a susceptible host[2,3]. STM relies on the acquisition of nutrients from macrophages for rapid intracellular replication[4,5]. As the microenvironment inside the macrophages is nutrient limited, STM must adapt and assimilate available nutrients within the macrophages to meet its nutritional needs[6]. It has been indicated that STM need to simultaneously exploit multiple nutrients, including glucose, glycerol, fatty acids, N-acetylglucosamine (GlcNAc), and several other carbon substances, in macrophages to sustain intracellular growth and cause systemic infection[4].

Recognition of microbial ligands by macrophages induces a shift in glucose metabolism from oxidative phosphorylation (OXPHOS) to aerobic glycolysis. This metabolic reprogramming in activated macrophages is referred to as the "Warburg effect", in which glucose is mainly converted to lactate via glycolysis and lactic acid fermentation even under normoxic conditions[7]. This increase in glycolysis is essential for the antibacterial functions of activated macrophages, as it evokes rapid ATP production to fuel inflammatory responses and cytokine production to control pathogen replication[7,8]. In addition, glycolysis supports the generation of reactive oxygen and nitrogen species, itaconate, and prostaglandins, all of which contribute to macrophage antimicrobial activities[9,10]. Furthermore, aerobic glycolysis in macrophages stimulated with bacterial lipopolysaccharide (LPS) is accompanied by upregulation of the serine synthesis pathway through the glycolytic intermediate 3-phosphoglycerate (3PG)[11–13]. The synthesis and downstream metabolism of serine is required for the optimal production of inflammatory cytokine interleukin (IL)-1β in activated macrophages[13]. IL-1β plays a major role in the induction and regulation of host defense[14], underscoring the importance of increase in serine synthesis to combat bacterial infection. The effects of macrophage metabolic changes on STM replication and pathogenicity are unknown.

Several intracellular bacterial pathogens have been shown to induce metabolic alterations in macrophages in a targeted, specific manner that benefit the pathogen[15]. Legionella pneumophila promotes increased glycolysis and decreased OXPHOS in infected macrophages, which is essential for L. pneumophila growth[16]. Francisella tularensis prevents the shift of infected macrophages to aerobic glycolysis to promote bacterial survival[17]. However, the mechanisms about how the altered macrophage metabolism promote these pathogens growth and how STM reprograms macrophage metabolism are largely unclear.

The STM genome encodes two distinct type III secretion systems (T3SSs) within large regions termed Salmonella pathogenicity island (SPI)-1 and SPI-2[18], which are two major determinants of STM pathogenesis. SPI-1 genes are mainly expressed during STM invasion of nonphagocytic epithelial cells, and SPI-1 effectors are required for epithelial cell invasion via triggering actin-dependent membrane ruffling[19]. SPI-2 genes are mainly expressed when the bacterium is inside the macrophage phagosome, and SPI-2 effectors interfere with host signaling pathways to convert the antimicrobial phagosome into the Salmonella-containing vacuole (SCV), where STM resides and replicates[20]. SPI-2 effectors also enable STM to more efficiently acquire host nutrients by inducing formation of tubular structures that extend from the SCV[5]. Several SPI-1 effectors are also expressed during STM resides within the SCV and contribute to intracellular replication, including SopE2, which is a guanidine exchange factor with the Rho GTPase Cdc42 as its primary target[21,22]. However, the molecular mechanism by which SopE2 promotes STM intracellular growth is currently unknown.

Here, we show that STM increases macrophage glycolysis and uses the SPI-1 effector SopE2 to repress macrophage serine synthesis, leading to accumulation of the macrophage glycolytic intermediates 2-phosphoglycerate (2PG), 3PG, and phosphoenolpyruvate (PEP). We show that STM uses a sophisticated regulatory pathway to detect low-glucose levels in infected macrophages and to induce uptake of these three accumulated glycolytic intermediates as carbon sources, thus promoting replication and virulence. In addition, STM-infected macrophages accumulate pyruvate and lactate, and STM senses these intermediates and subsequently upregulates SPI-2 gene expression through the CreBC two-component system, which promotes STM intracellular replication and systemic infection. Collectively, our findings indicate that STM induces a specific metabolic program that promotes the accumulation of glycolytic intermediates in macrophages, and that STM uses the accumulated glycolytic intermediates as carbon sources and cues to support its intracellular replication and pathogenicity.

## Results

**STM infection enhances glycolysis and reduces serine synthesis in infected macrophages.** Although STM has been previously shown to utilize glucose in macrophages[23], the effects of STM infection on macrophage glucose metabolism remain unclear. To determine whether STM infection of host macrophages leads to changes in macrophage glucose metabolism, we infected mouse peritoneal macrophages (PMs) with STM and assessed changes in macrophage glycolytic capacity at 2, 8, and 20 h post infection (hpi) with a Seahorse extracellular flux analyzer. Glycolysis was increased, and OXPHOS was decreased following STM infection, as indicated by a significant increase in extracellular acidification rate (ECAR) (Fig. 1a, Supplementary Fig. 1a) and a significant decrease in basal oxygen consumption rate (OCR) (Fig. 1b, Supplementary Fig. 1b). STM-infected PMs showed a significantly increased lactate production at 2, 8, and 20 hpi (Supplementary Fig. 1c), confirming that glycolysis was increased. Moreover, STM-infected PMs exhibited a significantly increased glucose uptake rate at 2, 8, and 20 hpi compared with uninfected PMs (Supplementary Fig. 1d), as also noted in a previous study in which STM infection led to increased glucose uptake by macrophages at 5 hpi[24], whereas the glucose uptake rate at 20 hpi is similar to that of 8 hpi, and both are higher compared with that of 2 hpi in STM-infected PMs (Supplementary Fig. 1d).

To define the specific changes in glycolytic flux that were induced by STM infection, we then conducted targeted quantitative metabolomics analysis, focusing specifically on STM-induced changes in central carbon metabolism. At 8 hpi, during active STM replication while the death of infected PMs is not evident (Supplementary Fig. 1e), STM-infected PMs showed significant increases of the flux through the glycolytic pathway (Fig. 1c, d, Supplementary Table 1). Following STM infection, we identified significant accumulations of glycolytic intermediates, including glucose 6-phosphate (G6P)/fructose 6-phosphate (F6P), fructose-1,6-bisphosphate (FBP), glyceraldehyde-3-phosphate (G3P), dihydroxyacetone phosphate (DHAP), phosphoglycerates (i.e., 2PG and 3PG), PEP, pyruvate, and lactate (Fig. 1c, d, Supplementary Table 1). The levels of some tricarboxylic acid (TCA) cycle intermediates, including citrate/isocitrate and malate were significantly decreased, while the levels of fumarate, succinate, and α-Ketoglutaric acid (α-KG) were significantly increased (Fig. 1c, d, Supplementary Table 1). The levels of the

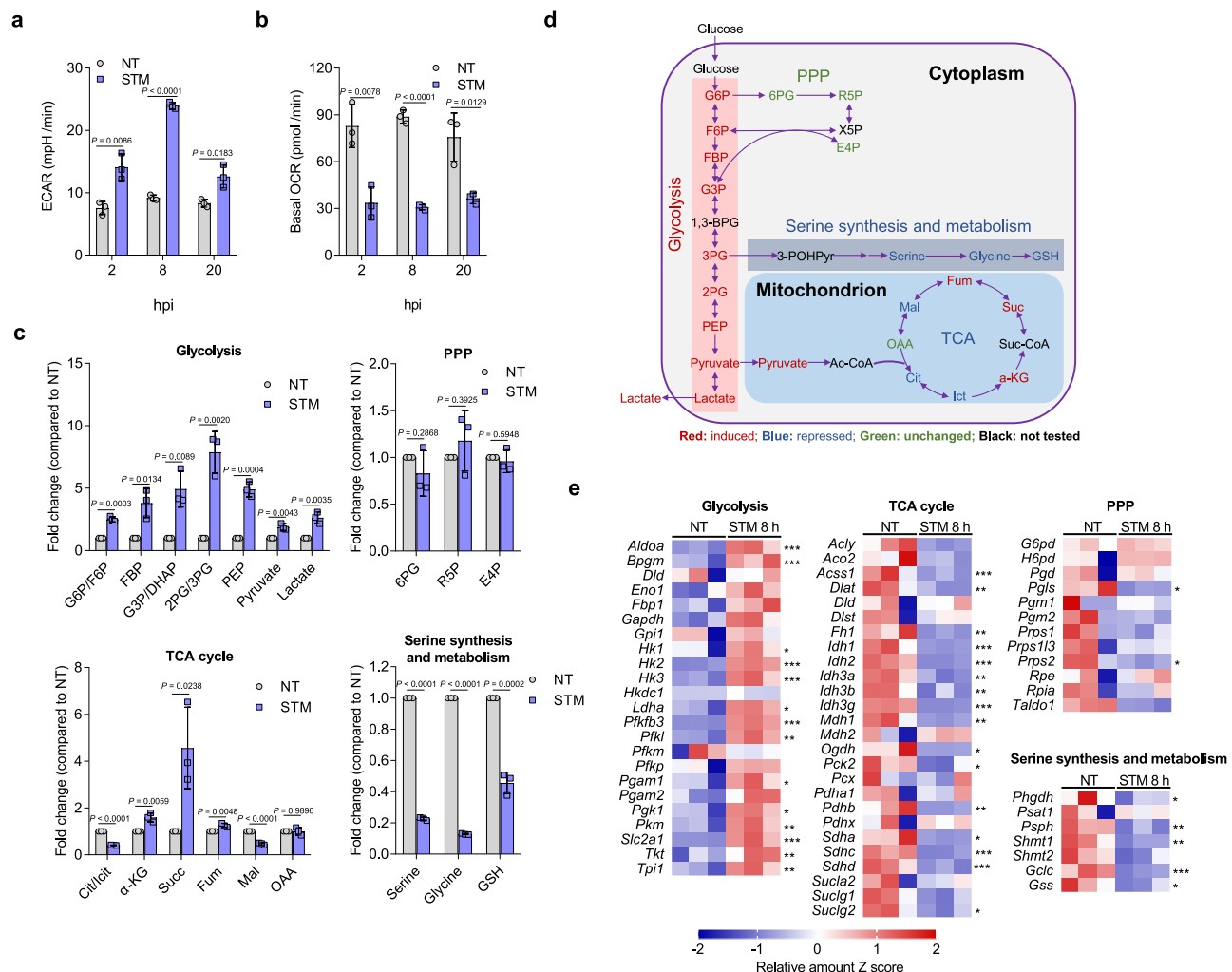

**Fig. 1** *Salmonella enterica* **serovar Typhimurium (STM) infection enhances glycolysis and reduces serine synthesis in infected macrophages. a, b,** Extracellular acidification rate (ECAR) (**a**) and basal oxygen consumption rate (OCR) (**b**) in untreated (NT) peritoneal macrophages (PMs) or those infected with STM for 2, 8, or 20 h. **c, d** Fold changes in glucose metabolites (**c**) and schematic of glucose metabolism (**d**) in STM-infected (8 h) PMs versus untreated PMs. Metabolites that were significantly enriched (red), diminished (blue), or not significantly changed (green) are indicated (**d**). glucose 6-phosphate (G6P), fructose 6-phosphate (F6P), fructose-1,6-diphosphate (FBP), glyceraldehyde-3-phosphate (G3P), dihydroxyacetophenone (DHAP), 3-phosphoglycerate (3PG), 2-phosphoglycerate (2PG), phosphoenolpyruvate (PEP), acetyl-CoA (Ac-CoA), citratre (Cit), isocitrate (Ict), α-ketoglutarate(α-KG), succinate (Suc), succinyl coenzyme A (Suc-CoA), fumarate (Fum), malate (Mal), oxaloacetate (OAA), 6-phosphogluconate (6PG), Ribose 5-phosphate (R5P), erythrose 4-phosphate (E4P), xylulose 5-phosphate (X5P), glutathione (GSH). Data are presented as mean ± SD, $n = 3$ independent experiments (**a–c**). **e** Heatmap of the expression profiles of glucose metabolism genes in STM-infected (8 h) PMs versus untreated PMs. $Z$ scores of the relative gene expression levels are displayed in the heatmaps ($n = 3$ independent experiments), with red representing higher and blue representing lower abundance. $P$ values were determined using one-way ANOVA (**a–c, e**). *, **, *** $P < 0.05, 0.01, 0.001$, respectively **e**. Source data are included in Source Data file.

tested PPP intermediates, 6-phosphogluconate (6PG), ribose 5-phosphate (R5P), and erythrose 4-phosphate (E4P), were not significantly changed (Fig. 1c, d, Supplementary Table 1). The levels of serine and its downstream metabolites glycine and glutathione (GSH) were significantly decreased in STM-infected PMs (Fig. 1c, d, Supplementary Table 1). These observed metabolic changes were dominated by host metabolites, as each STM cells accounted for 0.01 to 0.02% of the total metabolite concentrations (Supplementary Table 1) and ~94% of the infected PMs contained no more than ten bacteria at 8 hpi (Supplementary Fig. 1e). Together, the metabolomics data indicate that STM infection enhances glycolysis and reduces the synthesis and downstream metabolism of serine in infected macrophages.

Transcriptome analysis was conducted to extend the metabolomics findings, revealing that the gene expression profiles of STM-infected PMs were consistent with the observed metabolomic changes (Fig. 1e). The mRNA levels of key glycolytic enzymes, including hexokinase (Hk), 6-phosphofructo-2-kinase/fructose-2,6-bisphosphatase 3 (Pfkfb3), and dominant glucose transporter Slc2a1, were significantly upregulated in infected PMs relative to untreated PMs (Fig. 1e), which are consistent with the observed increase in the levels of glycolysis and glucose uptake of STM-infected PMs. The observed decreases in the levels of some TCA cycle intermediates can be explained by decreased mRNA expression of ATP citrate synthase (Acly), isocitrate dehydrogenase (Idh), α-KG dehydrogenase (Ogdh), and succinate dehydrogenase (Sdh) (Fig. 1e). The observed increases in the levels of α-KG, succinate, and fumarate could be attributed to the glutamine-dependent anerplerosis, which is activated during macrophages shift to a Warburg metabolism[25,26]. In this process,

glutamine is utilized as an anaplerotic carbon source to replenish TCA cycle intermediates, and α-KG is converted from glutamine to enter into the TCA cycle, resulting in the increase in the levels of succinate and fumarate[25]. Corresponding to the unchanged PPP metabolite levels, the mRNA levels of key PPP enzymes were not significantly changed (Fig. 1e). Furthermore, in accordance with the diminished serine synthesis pathway metabolite levels, the mRNA level of 3-phosphoglycerate dehydrogenase (Phgdh), the key enzyme in the serine synthesis pathway[27], and those of serine hydroxymethyltransferase 1 (Shmt1) and glutamate–cysteine ligase (Gclc), the key enzymes involved in glycine and GSH synthesis from serine[28,29], respectively, were significantly decreased (Fig. 1e). Collectively, these findings suggest that glycolytic metabolism is increased, and serine synthesis and downstream metabolic pathways were downregulated in STM-infected macrophages.

**Increased glycolysis in macrophages promotes STM intracellular replication.** We then investigated whether the observed increase in macrophage glycolysis is associated with STM intracellular replication. To inhibit macrophage glycolytic activities, we used a small interfering RNA (siRNA) to decrease the expression of the key glycolytic enzyme Pfkfb3 in the murine RAW264.7 macrophage cell line (48% reduction; Supplementary Fig. 2a). The knockdown significantly decreased STM replication in RAW264.7 cells (Fig. 2a), whereas pyruvate and lactate production was significantly decreased confirming that the glycolysis was inhibited in RAW264.7 cells owing to the knockdown (Supplementary Fig. 2b). Moreover, treatment of PMs with the glycolysis inhibitor PFK-15, which specifically inhibits Pfkfb3 and did not affect STM growth in LB medium and PRMI medium (Supplementary Fig. 2c), significantly decreased STM replication in PMs (Fig. 2b). These data indicate that increased glycolysis in macrophages during STM infection creates an intracellular environment conducive to STM replication.

**STM actively suppresses serine synthesis and increases 3PG levels in macrophages.** As STM infection reduced serine synthesis and downstream metabolism in PMs, we hypothesized that STM specifically represses serine synthesis to induce accumulation of the glycolytic intermediate 3PG, the precursor for de novo serine synthesis. To test this hypothesis, we infected PMs with live or heat-killed STM for 8 h in serine-depleted medium and measured the intracellular serine and 3PG concentrations in PMs. Compared with uninfected PMs, live STM-infected PMs showed significantly decreased serine levels, whereas heat-killed STM-infected PMs showed significantly increased serine levels (Fig. 2c, Supplementary Table 2). Notably, heat-killed STM induced significantly less 3PG accumulation in PMs than did live STM (Fig. 2c, Supplementary Table 2). In accordance with their differing serine levels, real-time quantitative PCR (RT-qPCR) analysis showed that *phgdh* transcription was significantly higher in PMs exposed to heat-killed STM than in those exposed to live STM (Fig. 2d).

Next, we overexpressed *phgdh* gene in RAW264.7 cells using the eukaryotic expression vector pcDNA3.1. *Phgdh* overexpression (2.0-fold increase; Supplementary Fig. 2d, left) resulted in significantly increased serine levels but decreased 3PG levels in live STM-infected RAW264.7 cells (Fig. 2e). In contrast, siRNA knockdown of Phgdh (39% reduction; Supplementary Fig. 2e, left) resulted in significantly decreased serine levels but increased 3PG levels in live STM-infected RAW264.7 cells (Fig. 2f). These data confirm that STM actively repressed macrophage serine synthesis, resulting in increased 3PG levels in infected macrophages.

**Suppressed serine synthesis in macrophages promotes STM intracellular replication.** We then investigated whether the reduction in serine synthesis in infected macrophages promotes intracellular STM replication. First, overexpression of *phgdh* in RAW264.7 cells (2.2-fold increase; Supplementary Fig. 2d, middle) significantly decreased STM intracellular replication (Fig. 2g). Second, siRNA-mediated Phgdh knockdown (44% reduction; Supplementary Fig. 2e, right) significantly increased STM replication in RAW264.7 cells (Fig. 2h). Finally, treatment with NCT-503, a *phgdh* inhibitor, significantly increased STM replication in PMs (Fig. 2i). Together, these results indicate that STM requires suppressed macrophage serine synthesis for maximal replication in macrophages.

**Uptake of macrophage-derived 3PG is essential for STM intracellular replication and systemic virulence.** We then investigated the mechanism by which macrophage glucose metabolism reprogramming promotes STM intracellular replication. Although glucose is an important carbon source during STM growth in macrophages[23], the availability of individual nutrients, including glucose, is limited in macrophages[4,30], leading us to hypothesize that STM can use the glycolytic intermediates that accumulate in macrophages as a carbon source. To test this hypothesis, we first confirmed that STM can utilize each of the accumulated intermediates as a sole carbon source for growth in vitro (Supplementary Fig. 3a). We then separately knocked out the bacterial transport systems for 2PG/3PG/PEP (referred to as 3PG hereafter, as these three intermediates can be easily interconverted and are imported through the same transporter in STM), G6P/F6P/G3P, pyruvate, and lactate, namely, PgtP[31,32], Uhpt[33], CstA-YbdD[34], and LldP[35], respectively. As expected, the mutant strains were unable to use the corresponding intermediates as a carbon source in vitro (Supplementary Fig. 3a). Although no specific transporters for FBP and DHAP have been reported, we showed that when FBP or DHAP were used as the sole carbon sources, in vitro growth of the *uhpt* mutant was inhibited (Supplementary Fig. 3a), indicating that Uhpt can transport both FBP and DHAP.

Next, as a previously published transcriptomic study showed that STM *pgtP*, *cstA*, *ybdD*, and *lldP* gene transcription was significantly upregulated in macrophages at 24 hpi but *uhpt* transcription was not significantly changed[36], we evaluated the influence of *pgtP*, *cstA-ybdD*, or *lldP* knockout on STM intracellular replication and pathogenicity. *pgtP* mutation significantly decreased STM replication in PMs, as revealed by the results of gentamicin protection assays (Fig. 3a) and immunofluorescence (Supplementary Fig. 3b, c). The 3PG and lactate levels showed no significant difference between *pgtP* mutant- and wild-type (wt)-infected PMs (Supplementary Fig. 3d), indicating PgtP does not influence macrophage 3PG accumulation and thus indicating the replication defect of the *pgtP* mutant in macrophages is owing to its inability to utilize 3PG. Furthermore, mice infected with the *pgtP* mutant exhibited a significantly increased survival rate and decreased bacterial burdens in the liver and spleen compared with mice infected with wt STM (Fig. 3b, c). *pgtP* mutant complementation with the *pgtP* gene restored intracellular replication and virulence in mice to wt levels (Fig. 3a–c). However, the intracellular growth and virulence in mice of the *cstA-ybdD* mutant, *lldP* mutant, and *cstA-ybdD lldP* double mutant did not differ from those of wt STM (Supplementary Fig. 3e–g), revealing the utilization of host pyruvate and lactate is not essential for STM intracellular replication and pathogenicity. These data indicate that PgtP is required for STM intracellular replication and systemic virulence and that, of the accumulated host glycolytic intermediates, 3PG uptake is

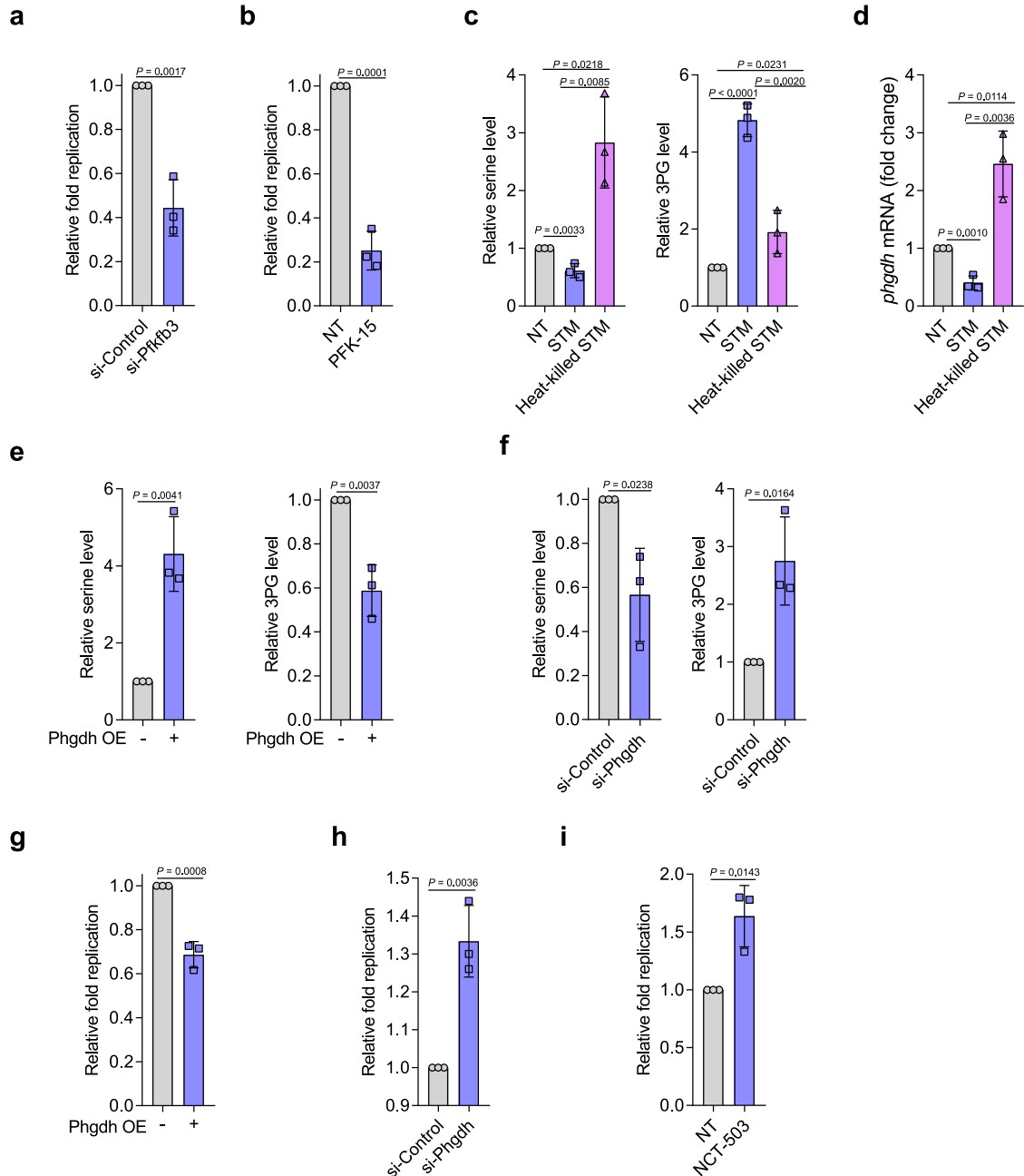

**Fig. 2 STM enhances glycolysis and reduces serine synthesis in macrophages to promote intracellular replication. a** Replication of STM in Pfkfb3 siRNA-treated or control siRNA-treated RAW264.7 cells. **b** Replication of STM in PFK-15-treated or untreated PMs. **c** Serine and 3-phosphoglycerate (3PG) levels in PMs infected with live or heat-killed STM for 8 h in serine-depleted RPMI medium. **d** *phgdh* mRNA levels in PMs infected with live or heat-killed STM for 8 h in serine-depleted RPMI medium. **e** Serine and 3PG levels in *phgdh*-overexpressing (OE:+) or control (OE:−) RAW264.7 cells cultured in serine-depleted RPMI medium. **f** Serine and 3PG levels in Phgdh siRNA-treated or control siRNA-treated RAW264.7 cells cultured in serine-depleted RPMI medium. **g** Replication of STM in *phgdh*-overexpressing or control RAW264.7 cells. **h** Replication of STM in Phgdh siRNA-treated or control siRNA-treated RAW264.7 cells. **i** Replication of STM in 5 μM NCT-503-treated or untreated PMs. Data are presented as mean ± SD, $n = 3$ independent experiments; $P$ values were determined using two-tailed unpaired Student's $t$ test (**a**, **b**, **d**–**i**) or one-way ANOVA (**c**). Source data are included in Source Data file.

essential for STM intracellular replication and systemic virulence in mice.

**Macrophage-derived 3PG is an important carbon source for STM intracellular replication.** Next, we considered that if STM exploits 3PG as a carbon source, the impaired replication and virulence of the *pgtP* mutant may be relieved by the addition of another carbon source during infection. Supporting this hypothesis, we observed that glucose supplementation resulted in

a significant increase in *pgtP* mutant replication in PMs (Fig. 3d), and that *pgtP* mutant infection resulted in significant higher mortality and greater bacterial burdens in the liver and spleen of glucose-treated mice than those of PBS-treated mice (Fig. 3e, f), demonstrating that 3PG was used by STM as a carbon source for intracellular growth. Furthermore, the reduced STM intracellular replication caused by macrophage serine synthesis upregulation was rescued by 3PG supplementation. When 3PG was added to the cell culture medium, STM intracellular replication in PMs and

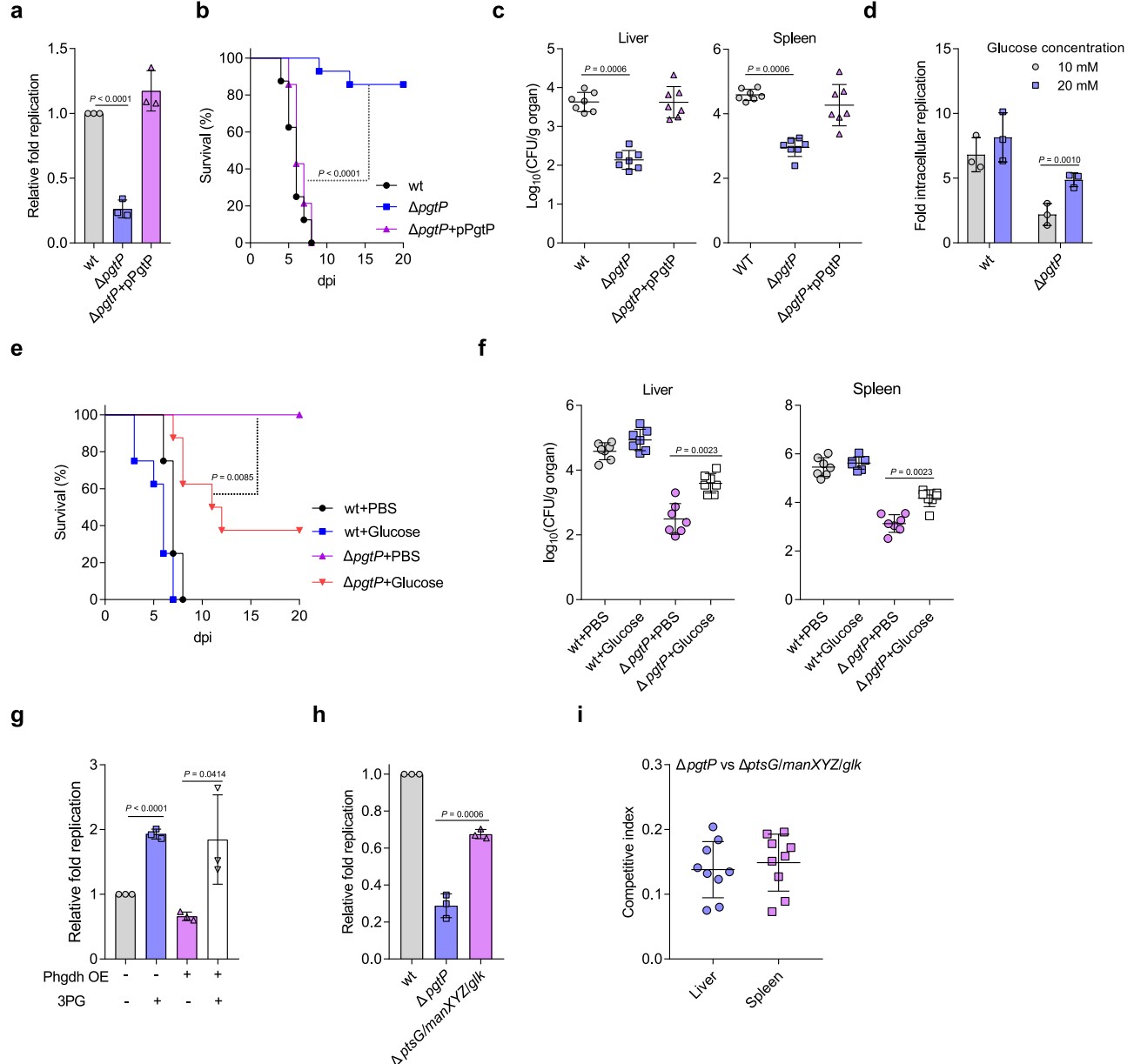

**Fig. 3 STM utilizes macrophage-derived 3PG as a major carbon source for intracellular replication and systemic infection. a** Replication of the wild-type (wt), *pgtP* mutant, or *pgtP* complemented STM strains in PMs. **b** Survival curves of mice infected i.p. with the wt (*n* = 16 mice), *pgtP* mutant (*n* = 14 mice), or *pgtP* complemented strains (*n* = 14 mice). **c** Liver and spleen bacterial burdens in mice infected with the wt, *pgtP* mutant, or *pgtP* complemented strains. **d** Replication of the wt or *pgtP* mutant STM strains in PMs cultured in RPMI medium containing 10 or 20 mM glucose. **e** Survival curves of mice infected i.p. with the wt or *pgtP* mutant STM strain and then injected i.p. with PBS or 20 mg of glucose every two days until 20 dpi or death (*n* = 8 mice per group). **f** Liver and spleen bacterial burdens in mice infected with the wt or *pgtP* mutant STM strain. Mice were infected as described in **e**. **g** Replication of wt STM in *phgdh*-overexpressing (OE+) or control (OE−) RAW264.7 cells in the presence or absence of 1 mM 3PG. **h** Replication of the wt, *pgtP* mutant, and *ptsG manXYZ glk* triple mutant STM strains in PMs. **i** Competitive index of the STM *pgtP* mutant versus the *ptsG manXYZ glk* triple mutant in the liver and spleen of infected mice. Data are presented as mean ± SD, *n* = 3 independent experiments **a**, **d**, **g**, **h**, *n* = 7 **c**, **f** or *n* = 9 **i** mice per group. *P* values were determined using two-tailed unpaired Student's *t* test (**a**, **g**, **h**), log-rank Mantel–Cox test (**b**, **e**), Mann–Whitney *U* test (**c**, **f**, **i**) or one-way ANOVA (**d**). *P* = 0.0002 and 0.0012 for liver and spleen, respectively, based on raw CFU values of the *pgtP* mutant vs. *ptsG manXYZ glk* triple mutant (**i**). Source data are included in Source Data file.

in *phgdh*-overexpressing RAW264.7 cells (2.7-fold increase in *phgdh* expression; Supplementary Fig. 2d, right) was significantly increased (Fig. 3g).

As glucose is thought to be required for the replication of STM in macrophages[23,30], next we compared the roles of glucose and 3PG in STM intracellular replication and systemic virulence. Compared with that of a *ptsG manXYZ glk* triple mutant, which can neither take up nor utilize glucose[23], the replication of the

*pgtP* mutant in PMs was significantly decreased (Fig. 3h). We then performed in vivo competition assays to measure the difference in systemic tissue colonization between *pgtP* mutant and *ptsG manXYZ glk* triple mutant. The result showed that *pgtP* mutant was significantly outcompeted by the *ptsG manXYZ glk* triple mutant in spleen and liver of the infected mice (Fig. 3i), implying the virulence of the *pgtP* mutant is more attenuated than that of the *ptsG manXYZ glk* triple mutant. These data

indicate that host-accumulated 3PG might play a more important role in STM intracellular replication and virulence than glucose.

**STM uses the SPI-1 effector SopE2 to repress macrophage serine synthesis**. The above results show that STM represses macrophage serine synthesis to trigger 3PG accumulation in macrophages, and that 3PG is used by intracellular STM as a major carbon source; thus, we next investigated the mechanisms by which STM represses macrophage serine synthesis. We first examined whether the effectors encoded by SPI-1 or SPI-2 influence *phgdh* gene expression. PMs infected with an SPI-1 mutant, but not those infected with an SPI-2 mutant, exhibited upregulated *phgdh* gene transcription compared with PMs infected with wt STM (Fig. 4a). The *sopE2* mutant did not repress *phgdh* transcription in PMs, but other SPI-1 effector gene mutants did (Fig. 4b), and complementing the *sopE2* mutant with *sopE2* repressed *phgdh* transcription and expression in PMs to a level comparable to that in wt STM-infected PMs (Fig. 4b, c). In addition, transcription of the two key glycolytic genes, *pfkfb3* and *slc2a1*, was unaffected by the mutation of *sopE2* (Supplementary Fig. 4a). *sopE2* overexpression in RAW264.7 cells decreased *phgdh* transcription and expression (Fig. 4d, e), whereas PMs infected with the *sopE2* mutant exhibited decreased intracellular 3PG levels and increased intracellular serine levels compared with PMs infected with wt STM (Fig. 4f). Finally, *sopE2* mutant replication in PMs was significantly reduced compared with wt STM replication in PMs (Fig. 4g), but was increased by exogenous 3PG addition (Fig. 4h). Together, these results indicate that SopE2 is the factor responsible for suppression of macrophage serine synthesis.

SopE2 is a guanine nucleotide exchange factor whose primary target is the host Rho GTPase Cdc42[37]. We thus sought to determine whether SopE2 represses *phgdh* expression through Cdc42. siRNA knockdown of Cdc42 (70% reduction; Fig. 4i) significantly increased *phgdh* gene transcription in HeLa cells (Fig. 4j), whereas both the intracellular 3PG level and intracellular bacterial replication were significantly reduced in Cdc42 siRNA-treated cells (Fig. 4k, l). Furthermore, addition of exogenous 3PG increased bacterial replication in Cdc42 siRNA-treated cells (Fig. 4m). These results indicate that STM represses the macrophage serine synthesis pathway through a SopE2-mediated Cdc42 mechanism.

We next tested the involvement of SopE2 in STM systemic infection. SopE2 is known to be required for STM to enter and cross the intestinal epithelium[38]. To bypass intestinal invasion and test the systemic infection of STM, mice were infected by the intraperitoneal route. Compared with mice infected with wt STM, mice infected with the *sopE2* mutant exhibited a significantly increased survival rate and reduced bacterial burdens in the liver and spleen (Supplementary Fig. 4b, c), indicating that SopE2 contributes to STM systemic infection. Because SopE2 and PgtP are involved in the macrophage accumulation and STM utilization of 3PG, respectively, we then compared the roles of SopE2 and PgtP in STM intracellular replication and systemic infection. We observed that the replication of the *pgtP* mutant in PMs was significantly decreased compared to that of the *sopE2* mutant, whereas comparable to that of the *sopE2 pgtP* double mutant (Supplementary Fig. 4d). The in vivo competition assays showed that the *pgtP* mutant was significantly outcompeted by the *sopE2* mutant, whereas competed equally with the *sopE2 pgtP* double mutant in liver and spleen of the infected mice (Supplementary Fig. 4e). Together, these data suggest that the contribution of SopE2 to STM systemic virulence is entirely dependent on PgtP, and the suppression of macrophage serine synthesis by SopE2 is necessary for STM systemic infection.

**STM senses decreased glucose levels in macrophages to upregulate bacterial 3PG uptake**. To reveal how STM takes advantage of the elevated 3PG levels in infected macrophages, we investigated the regulatory system involved in STM *pgtP* upregulation inside macrophages. Upon investigating the regulatory functions of 18 putative regulatory proteins that present in STM but absent in *Escherichia coli* (Supplementary Table 3) by transcriptome analysis, we identified STM2748 (named VrpA, virulence-related regulatory protein A) as a potential positive regulator of *pgtP*, as the transcription of *pgtP* and the other three genes in the *pgt* gene cluster was downregulated in the *vrpA* mutant (NCBI SRA accession: PRJNA561041). RT-qPCR analysis further confirmed the positive regulation of *pgtP* by VrpA when STM grown in N-minimal medium (Fig. 5a), which has a mildly acidic pH and low magnesium and phosphorus concentrations that recapitulate the conditions inside host macrophages. Consistent with the role of *pgtP* in STM intracellular replication, *vrpA* mutation significantly decreased STM replication in PMs (Fig. 5b).

Next, we tested the influence of *vrpA* mutation on STM virulence in mice. Compared with mice infected with wt STM, mice infected with the *vrpA* mutant exhibited a significantly increased survival rate and reduced bacterial burdens in the liver and spleen (Fig. 5c, d). Moreover, complementation of the *vrpA* mutant with *vrpA* restored bacterial replication in PMs and virulence in mice, whereas complementation of the *vrpA* mutant with *pgtP* partially restored bacterial replication in PMs and virulence in mice (Fig. 5b–d), further confirming that *pgtP* is controlled by *vrpA*. The above data indicate that the contribution of VrpA to STM virulence is partially dependent on PgtP, and this was further confirmed by the facts that the replication of the *vrpA pgtP* double mutant was significantly decreased in PMs compared with that of the *pgtP* mutant (Supplementary Fig. 5a) and *vrpA pgtP* double mutant was significantly outcompeted by the *pgtP* mutant in liver and spleen of the infected mice (Supplementary Fig. 5b).

We then investigated whether VrpA directly regulates *pgtP*. Electrophoretic mobility shift assays (EMSAs) and chromatin immunoprecipitation and quantitative PCR (ChIP-qPCR) assays revealed that VrpA binds to the *pgtP* promoter both in vitro and in vivo (Fig. 5e, f), indicating that VrpA activates *pgtP* directly.

To determine how VrpA is induced when STM inside macrophages, the upstream regulatory system(s) were then investigated. Using program BPROM[39], we identified a putative CRP-binding site in the *vrpA* promoter region, indicating CRP might be a potential regulator of VrpA. *crp* mutation significantly decreased *vrpA* and *pgtP* transcription levels (Fig. 5g), implying that CRP positively regulates *vrpA* and *pgtP* expression. Consistent with the roles of *vrpA* and *pgtP* in STM intracellular replication and systemic infection, *crp* mutation significantly decreased STM replication in PMs (Supplementary Fig. 5c); mice infected with a *crp* mutant exhibited a markedly increased survival rate and reduced bacterial burdens in the liver and spleen (Supplementary Fig. 5d, e).

The results of EMSA and ChIP-qPCR analyses showed that CRP binds to the *vrpA* promoter both in vitro and in vivo (Fig. 5h, i), confirming that CRP directly regulates *vrpA*. In addition, CRP regulates its target genes upon complexation with cyclic adenosine monophosphate (cAMP), and its regulatory activity is determined by the cAMP level;[40,41] we found that cAMP was also required for CRP binding to the *vrpA* promoter in vitro (Fig. 5h). Together, these results indicate that CRP is a direct regulator of VrpA in STM, which regulates *pgtP* during intracellular replication.

As the availability of glucose inside macrophages is limited during bacterial infection[4], we hypothesized that decreased

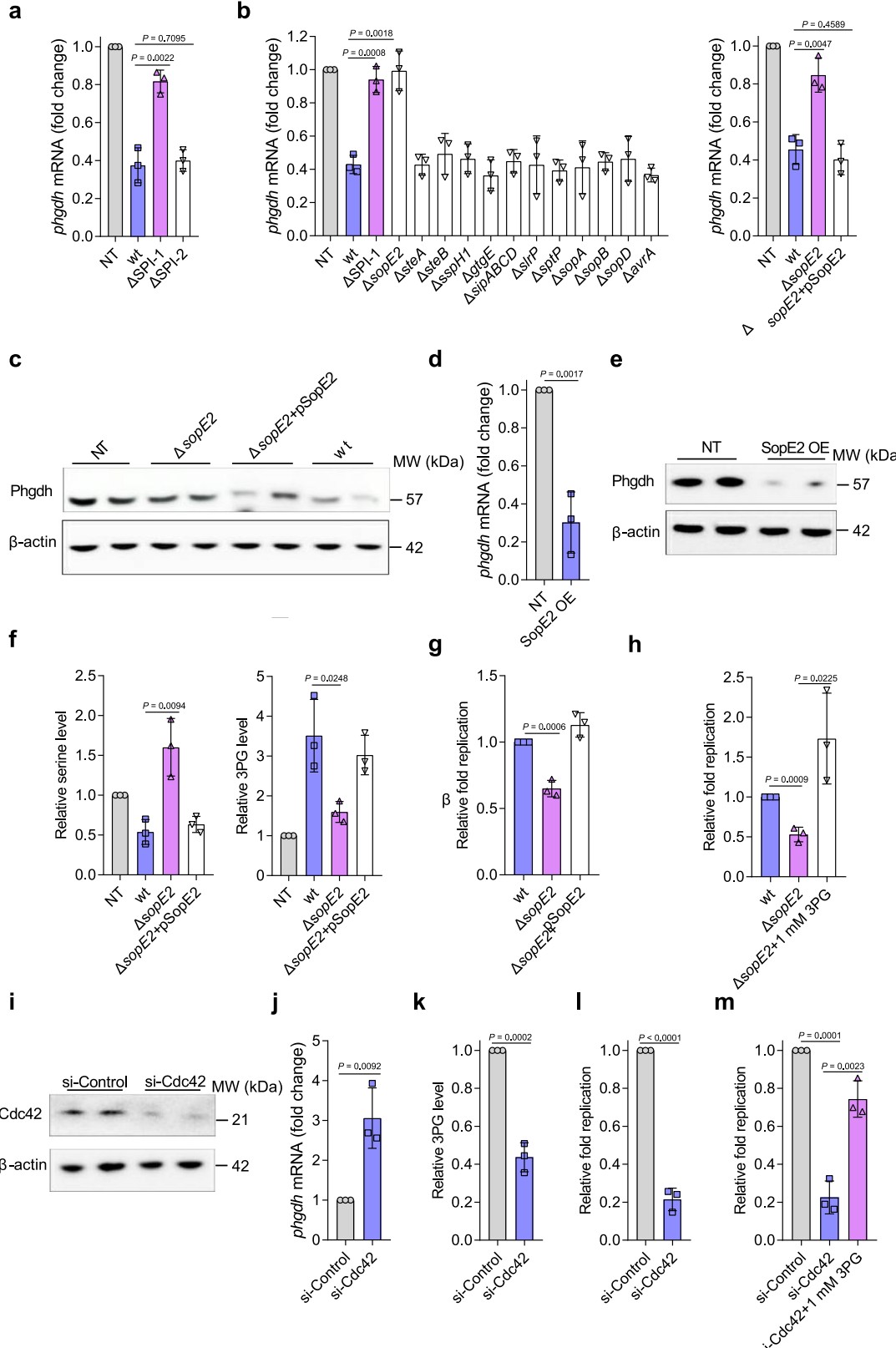

glucose uptake by the STM phosphotransferase (PTS) system increases cAMP levels and leads to *crp* upregulation in STM. We next investigated cAMP levels and *crp* expression in intracellular STM during infection, and found both to increase (Fig. 5j, k). Consistent with these increases, *vrpA* and *pgtP* transcription levels in intracellular STM also increased (Fig. 5k).

We then investigated whether *vrpA*, *pgtP*, and *crp* expression was negatively regulated by glucose availability inside macrophages. To this end, we cultured PMs with Roswell Park Memorial Institute Medium (RPMI) medium containing 0.25, 0.5, or 2 g/L glucose for 16 h before infection with STM. Intracellular *vrpA*, *pgtP,* and *crp* transcription was induced earlier

**Fig. 4 STM uses the *Salmonella* pathogenicity island (SPI)-1 effector SopE2 to repress macrophage serine synthesis. a** Quantitative PCR (RT-qPCR) analysis of *phgdh* mRNA levels in untreated (NT) PMs or PMs infected with wt STM, SPI-1 mutant, or SPI-2 mutant for 8 h. **b** RT-qPCR analysis of *phgdh* mRNA levels in untreated PMs or those infected with wt STM, the indicated mutant or complemented strain for 8 h. **c** Immunoblot analysis of Phgdh protein levels in untreated PMs or those infected with the wt, *sopE2* mutant, or *sopE2*-complemented strain for 8 h. **d** RT-qPCR analysis of *phgdh* mRNA levels in *sopE2*-overexpressing (SopE2 OE) or control RAW264.7 cells. **e** Immunoblot analysis of Phgdh protein levels in *sopE2*-overexpressing or control RAW264.7 cells. **f** Serine and 3PG levels in PMs infected with wt STM or *sopE2* mutant for 8 h. **g** Replication assays of the wt, *sopE2* mutant, or *sopE2*-complemented strains in PMs in the presence or absence of 1 mM 3PG. **h** Replication assays of the wt or *sopE2* mutant in PMs in the presence or absence of 1 mM 3PG. **i** Immunoblot analysis of Cdc42 protein levels, **j** RT-qPCR analysis of *phgdh* mRNA levels, **k** 3PG levels, **l** replication of wt STM, and **m** replication of wt STM in the presence or absence of 1 mM 3PG, in Cdc42 siRNA-treated or control siRNA-treated HeLa cells. Data are presented as mean ± SD, *n* = 3 independent experiments; *P* values were determined using two-tailed unpaired Student's *t* test (**a**, **b**, **d**, **f–h**, **j–m**). Immunoblots are representative of three independent experiments (**c**, **e**, **i**). Source data are included in Source Data file.

and at a higher level in PMs cultured in RPMI medium containing low-glucose concentrations (0.5 g/L or 0.25 g/L) than in PMs cultured with RPMI medium containing a high glucose concentration (2 g/L) (Supplementary Fig. 5f). It is highly likely that lower glucose concentration in the medium resulted in its lower concentration within macrophages, and thus bacterial *vrpA*, *pgtP*, and *crp* transcription was induced earlier and at a higher level. Moreover, when STM was cultured in RPMI medium containing different glucose concentrations (0–4 g/L), the transcription of these genes was more highly induced in bacteria cultured with no-glucose or low-glucose concentrations than in those cultured with high glucose concentrations (Supplementary Fig. 5g). We also repressed cellular glycolysis via Pfkfb3 knockdown in RAW264.7 cells before STM infection, and found that *crp*, *vrpA*, and *pgtP* transcription of the intracellular STM was significantly decreased upon Pfkfb3 knockdown (Supplementary Fig. 5h), further confirmed the induction of *vrpA*, *pgtP*, and *crp* expression in response to glucose limitation. Collectively, these data indicate that during STM infection of macrophages, the bacterial cAMP-CRP content increases owing to decreased glucose levels in macrophages, and that cAMP-CRP activates *pgtP* expression through VrpA, facilitating 3GP uptake.

**STM senses macrophage-accumulated pyruvate and lactate by the two-component system CreBC to activate SPI-2 genes.** Although our results indicate that the accumulated macrophage pyruvate and lactate were not utilized by STM as nutrients for intracellular replication, it is possible that they act as cues to control STM gene expression. Consistent with this hypothesis, the transcription of STM *creB* and *creC* genes, which encode the two-component system CreBC that is responsive to pyruvate and lactate[42], was upregulated in macrophages at 24 hpi[36]. To determine whether the protein levels of CreB and CreC are increased inside macrophages, we constructed a *creBC* promoter-*lux* fusion (*creBC-lux*) and then compared its luminescence intensity inside PMs at 2, 8, and 20 hpi to that in RPMI medium. The results showed that the luminescence intensity of *creBC-lux* inside PMs was significantly increased at each time point post infection (Supplementary Fig. 6a), confirming the increased expression of CreBC during STM growth in macrophages. The upregulation of STM *creBC* inside macrophages indicates that CreBC might contribute to STM intracellular replication. Therefore, we knocked out the regulator *creB* and then tested the influence of *creB* knockout on STM intracellular replication and pathogenicity. We found that *creB* mutation significantly decreased STM replication in PMs, as revealed by the gentamicin protection assays (Fig. 6a) and immunofluorescence (Supplementary Fig. 3b, c). Moreover, mice infected with the *creB* mutant exhibited a significantly increased survival rate and decreased bacterial burdens in the liver and spleen (Fig. 6b, c). *creB* mutant complementation with the *creB* gene restored bacterial intracellular replication and virulence to WT levels both in vitro and

in vivo (Fig. 6a–c). Together, these data suggest that CreBC is essential for STM intracellular replication and infection in mice.

We then confirmed that pyruvate and lactate accumulation in macrophages induces *creBC* expression in STM. First, *creB* and *creC* transcription levels were significantly increased by the addition of lactate and pyruvate in vitro (Fig. 6d). Second, *creB* and *creC* transcription levels were significantly increased in bacteria cultured with lysate from STM-infected cells compared with bacteria cultured with lysate from mock-infected cells (Supplementary Fig. 6b). Third, we inhibited glycolysis and cellular pyruvate and lactate production via Pfkfb3 knockdown in RAW264.7 cells before STM infection, and found that *creB* transcription of the intracellular STM was significantly decreased upon Pfkfb3 knockdown (Supplementary Fig. 6c).

We next investigated the means by which pyruvate and lactate promote STM intracellular replication and virulence through CreB. RNA-seq data show that loss of *creB* reduced the transcription of all 33 SPI-2 genes in STMs cultured in N-minimal medium (Fig. 6e), and RT-qPCR analysis confirmed the positive regulation of SPI-2 gene expression by CreB in N-minimal medium (Supplementary Fig. 6d). In addition, *creB* mutation significantly decreased the transcription levels of the representative SPI-2 gene *ssaG* inside PMs at 2, 8, and 20 hpi, respectively (Supplementary Fig. 6e). Moreover, both pyruvate and lactate induced *ssaG* transcription in STM cultured in N-minimal medium, and *creB* mutation abolished this effect (Supplementary Fig. 6f), indicating that pyruvate and lactate induce SPI-2 gene expression in a CreB-dependent manner. However, *lldp* and *cstA-ybdD* double mutation did not influence *ssaG* transcription when STM grown in N-minimal medium supplied with pyruvate or lactate (Supplementary Fig. 6g), revealing that the utilization of pyruvate and lactate is not related with SPI-2 gene expression. Taken together, these data suggest that STM uses host-accumulated pyruvate and lactate as cues to activate SPI-2 gene expression via CreB.

**CreB activates SPI-2 genes through the regulator VrpB.** To further delineate the mechanism by which CreB contributes to STM intracellular replication and pathogenesis, we next investigated the pathway associated with SPI-2 regulation by CreB. Most previously identified SPI-2 regulators have been shown to activate SPI-2 gene expression through the regulators SsrA and/or SsrB[43]. However, our EMSA results showed that CreB did not bind to the promoter of either *ssrA* or *ssrB* (Supplementary Fig. 6h), suggesting that CreB may activate SPI-2 indirectly. ChIP-seq analysis showed that the promoter region of STM2180 (named *vrpB*), which encodes a LysR family transcriptional regulator, was significantly enriched in the CreB-ChIP sample compared with that in the control sample (NCBI SRA accession: PRJNA561041), suggesting that *vrpB* might be a direct CreB target. CreB binds a conserved cre tag sequence to regulate downstream genes in *Escherichia coli*[44], but the upstream region of the *vrpB* gene does not contain a typical cre tag sequence. Through EMSA and ChIP-

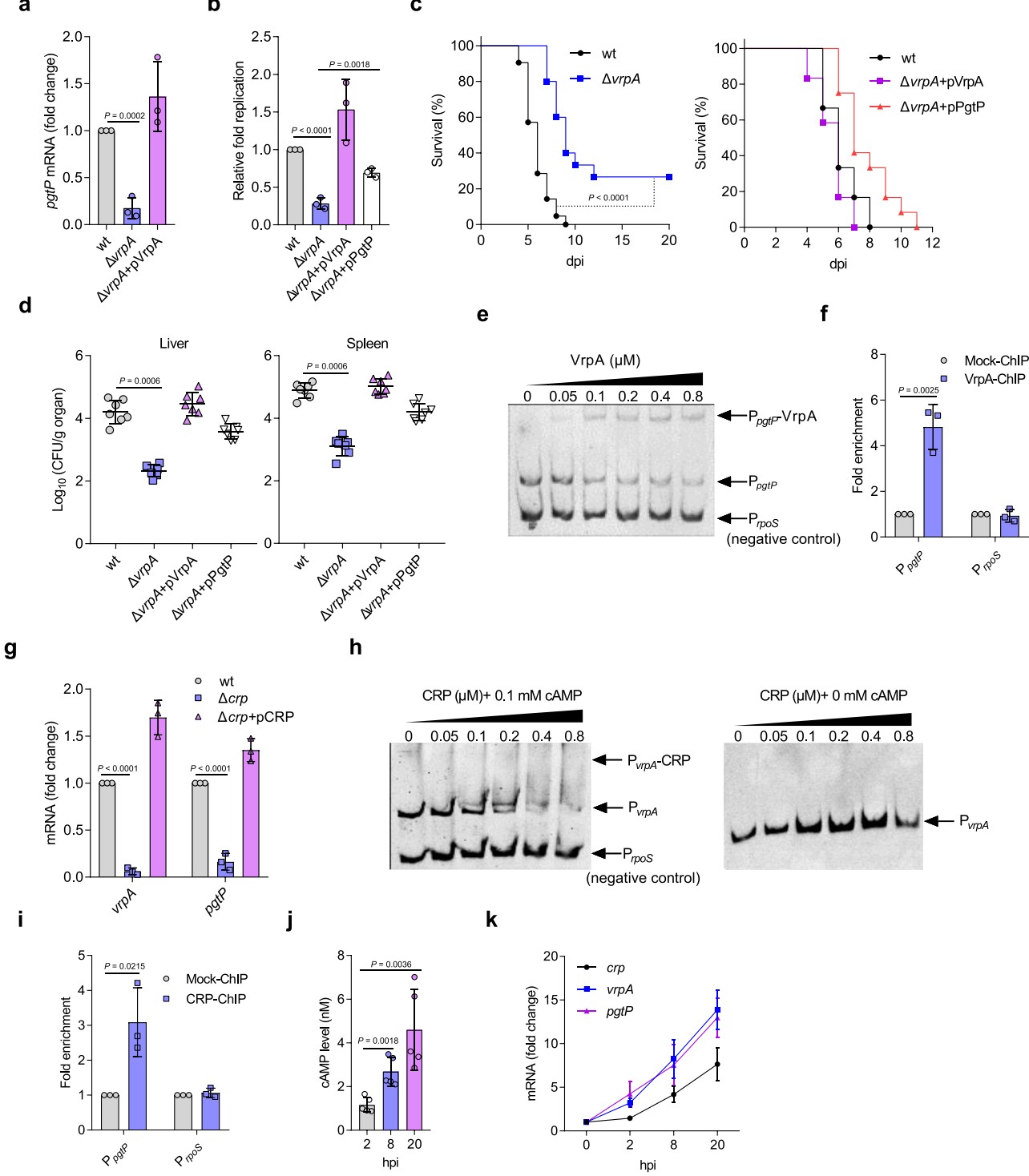

qPCR, we found that CreB bound to the *vrpB* promoter both in vitro and in vivo (Fig. 6f, g), confirming the direct regulation of *vrpB* by CreB. *creB* mutation significantly decreased *vrpB* gene transcription in STM cultured in N-minimal medium (Supplementary Fig. 6i). Together, these results reveal that CreB directly and positively regulates *vrpB*.

We then examined the regulatory mechanisms between VrpB and SPI-2 genes. Therefore, we knocked out *vrpB* and tested the influence of *vrpB* knockout on the expression of SPI-2 genes. *vrpB* mutation significantly decreased *ssaG* gene transcription in STM cultured in N-minimal medium (Supplementary Fig. 6j). EMSA

and ChIP-qPCR results revealed that VrpB binds to the *ssrA* and *ssrB* promoters, indicating that VrpB activates SPI-2 gene expression directly (Fig. 6h, i). Consistent with the roles of CreB and SPI-2 in intracellular replication and systemic infection, *vrpB* mutation significantly decreased STM replication in PMs (Fig. 6j). In addition, mice infected with the *vrpB* mutant exhibited a significantly increased survival rate and decreased bacterial burdens in the liver and spleen (Fig. 6k, l). The replication of the *creB vrpB* double mutant in PMs was significantly decreased compared to that of the *vrpB* mutant (Supplementary Fig. 6k) and *creB vrpB* double mutant was significantly outcompeted by the

**Fig. 5 STM senses decreased glucose levels in macrophages to upregulate bacterial 3PG uptake. a** RT-qPCR analysis of *pgtP* mRNA levels in the wt, *vrpA* mutant, and *vrpA*-complemented STM strains. Bacteria were grown in N-minimal medium for 6 h before collection. **b** Replication of the wt, *vrpA* mutant, and *vrpA*-complemented strains in PMs. **c** Survival curves of mice infected i.p. with the wt, *vrpA* mutant, or complemented strains. Left: *n* = 21 mice for wt, *n* = 15 mice for *vrpA* mutant. Right: *n* = 14 mice per group. **d** Liver and spleen bacterial burdens in mice infected with the wt, *vrpA* mutant, or complemented strains. **e** Electrophoretic mobility shift assay (EMSA) of the *pgtA* promoter with purified VrpA protein. **f** Fold enrichment of the *pgtA* promoter in VrpA-chromatin immunoprecipitation (ChIP) samples. **g** RT-qPCR analysis of *vrpA* and *pgtP* mRNA levels in the wt, *crp* mutant, and *crp*-complemented strains. Bacteria were grown in N-minimal medium for 6 h before collection. **h** EMSA of the *vrpA* promoter with purified CRP protein in the presence or absence of 0.1 mM cAMP. **i** Fold enrichment of the *vrpA* promoter in CRP-ChIP samples. **j** cAMP levels in STM collected from infected PMs at 2, 8, and 20 h post infection (hpi). **k** RT-qPCR analysis of *crp*, *vrpA*, and *pgtP* mRNA levels in intracellular STM. Fold changes in intracellular gene expression level at 2, 8, and 20 hpi relative to the expression of the corresponding genes in RPMI medium (0 hpi) are presented. Data are presented as mean ± SD, *n* = 3 (**a, b, f, g, i, k**) or *n* = 5 independent experiments (**j**), *n* = 7 mice per group (**d**). Images are representative of three independent experiments (**e, h**). *P* values were determined using two-tailed unpaired Student's *t* test (**a, b, j**), log-rank Mantel–Cox test (**c**), Mann–Whitney *U* test (**d**) or one-way ANOVA (**f, g, i**). Source data are included in Source Data file.

*vrpB* mutant in liver and spleen of the infected mice (Supplementary Fig. 6l), indicating that the contribution of CreB to STM virulence is partially dependent on VrpB. In contrast, the intracellular replication of the *vrpB ssrB* double mutant is comparable to that of the *ssrB* mutant (Supplementary Fig. 6m) and the *vrpB ssrB* double mutant competed equally with the *ssrB* mutant in liver and spleen of the infected mice (Supplementary Fig. 6n), indicating that the contribution of VrpB to STM virulence are entirely dependent on SPI-2. This was further confirmed by the fact that complementation of the *vrpB* mutant with *ssrB* restored STM intracellular replication and virulence (Fig. 6j–l). Collectively, these results confirm VrpB as a positive regulator of SPI-2 genes and indicate that CreB activates SPI-2 gene expression by directly activating VrpB.

Consistent with these suggested functions, *creB*, *vrpB*, and *ssaG* transcript levels increased during STM growth in PMs (Fig. 6m). In addition, the *creB*, *vrpB*, and *ssaG* transcript levels were upregulated by the addition of lactate and pyruvate in vitro, and either *creB* or *vrpB* mutation abolished *ssaG* upregulation by lactate and pyruvate (Fig. 6n), further confirming the *creB*–*vrpB*–SPI-2 regulatory pathway. Also, the transcription of *vrpB* and *ssaG* was significantly decreased when STM replicated in Pfkfb3 knockdown RAW264.7 cells (Supplementary Fig. 6c). Collectively, these results indicate that during STM infection, CreBC activates SPI-2 genes through VrpB in response to increased pyruvate and lactate levels in macrophages, thus promoting STM intracellular replication and virulence.

## Discussion

STM replication inside macrophages is a crucial step in the induction of systemic infection[2,3], and requires large amounts of nutrients[4,45]. We studied the metabolic reprogramming of macrophages during STM infection and revealed the mechanisms by which this active reprogramming supports STM intracellular replication and systemic virulence; notably, this STM-driven macrophage metabolic reprogramming differs from the host-driven reprogramming that occurs when macrophages are activated by heat-killed STM or LPS. We also showed that STM uses the glycolytic intermediates that accumulate in macrophages as carbon sources and cues to establish successful intracellular replication, thus enabling systemic infection (see Fig. 7 for a schematic of our proposed mechanism).

Previously, although several reports indicated that STM blocks host OXPHOS during infection of the mouse colon mucosa[46] and during growth in mouse bone marrow-derived macrophages[36], it was unclear whether STM induces a Warburg shift in host metabolism during intracellular infection. We found that STM does induce a Warburg shift in macrophages, characterized by increased glycolysis and decreased TCA cycle and OXPHOS activities. Through in vitro and in vivo infection assays, we

showed that this shift supports the effective intracellular replication and full virulence of STM.

Glucose uptake is upregulated in STM-infected macrophages, presumably to combat STM infection[24]. STM has several glucose transport systems and can directly access macrophage glucose to support rapid intracellular replication[23,24]. This competition for glucose in infected macrophages results in decreased glucose levels; however, STM can use diverse carbon sources in macrophages, as previously reported[4], and it further reprograms host metabolism, as observed in this study.

We showed that STM modulates macrophage glucose metabolism to promote the accumulation of 3PG in host macrophages, which STM can utilize as a carbon source. The observation that the STM mutant with defective 3PG uptake is less replicative in macrophages and exhibits more-attenuated virulence in mice than the mutant with defective glucose utilization implies that more 3PG than glucose is available in infected macrophages. However, STM might still preferentially use glucose, as STM starts to take up 3PG only when the glucose level in macrophages decreases. STM regulates the uptake of 3PG in response to its environment owing to a newly discovered regulatory pathway that senses low-glucose concentrations and upregulates 3PG uptake to support intracellular replication and virulence. This regulatory pathway contains a newly characterized regulator VrpA, which links the regulation of 3PG uptake gene *pgtP* to the classic bacterial cAMP-CRP regulatory system.

We further investigated the serine synthesis pathway downstream of glycolysis and found that macrophages activated by heat-killed STM upregulated the serine synthesis pathway, confirming that this effect is a host-driven metabolic shift in activated macrophages. Notably, increases in serine synthesis pathway activity are required for cytokine IL-1β production in activated macrophages[13], highlighting the importance of the upregulation of serine synthesis pathway to combat infection. However, we found that live intracellular STM represses the macrophage serine synthesis pathway via the SPI-1 effector SopE2, leading to increased levels of intracellular carbon sources that can be used by STM, and thus to an increase in STM intracellular replication. Serine is a central node in the biosynthesis of nucleotides, NADPH, and GSH[47], and all of these processes are related to the innate immune response of macrophages. Importantly, Phgdh regulates both mitochondrial and pentose phosphate metabolism[48], affecting the itaconate and succinate production and antibacterial activities of macrophages[49]. However, whether the decreased serine synthesis pathway activity and *phgdh* gene expression in STM-infected macrophages influence cellular innate immunity and antibacterial responses requires future study.

It was previously known that SopE2 contributes to both epithelial cell invasion and intracellular replication of *Salmonella*[22], but the molecular mechanism underlying the growth promotion

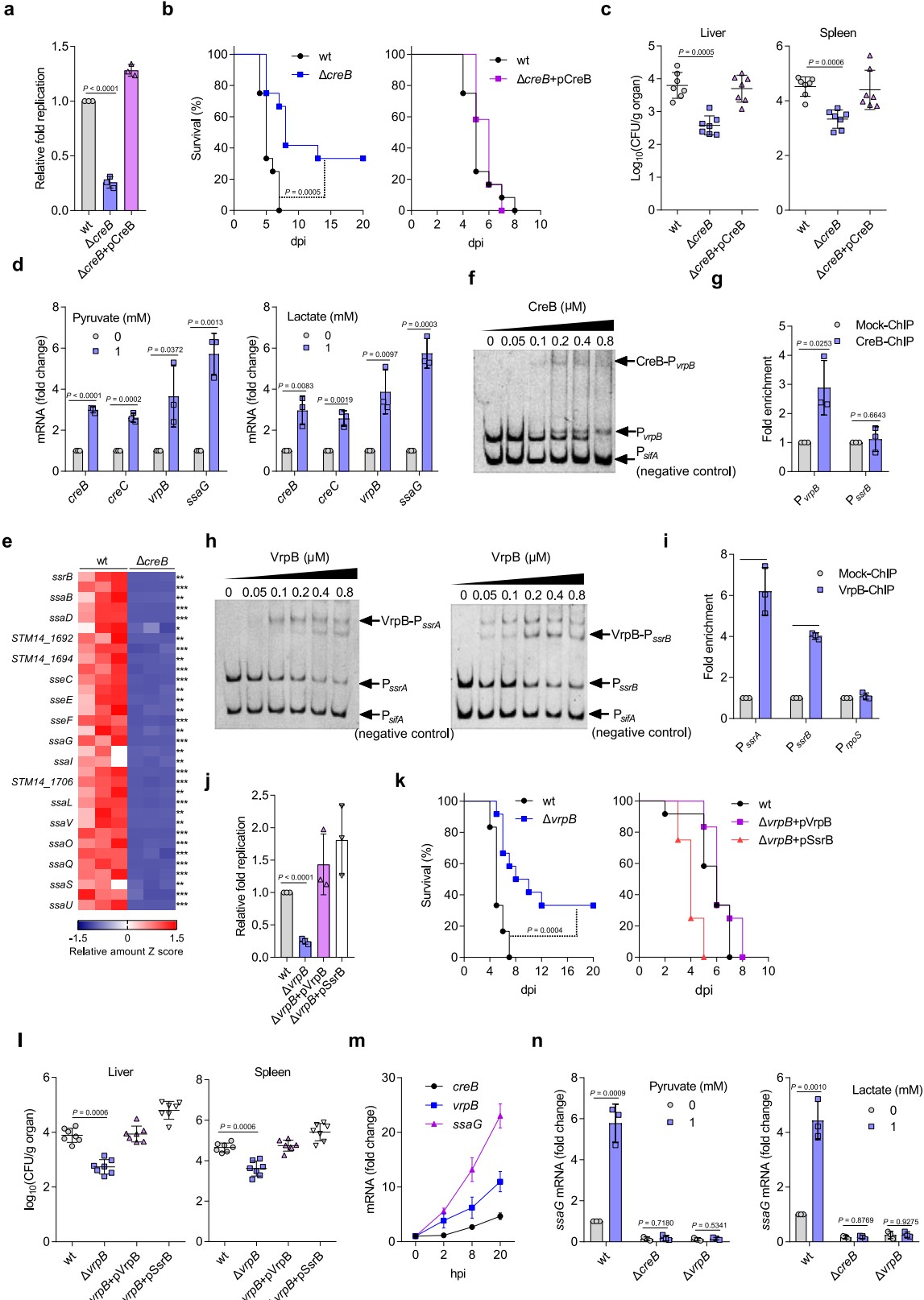

by SopE2 remains unknown. We found that SopE2 can repress macrophage *phgdh* gene expression via Cdc42 signaling, promoting the accumulation of intracellular glycolytic intermediates that can be used by STM as carbon sources. Therefore, our results provide a mechanism for how SopE2 involved in STM intracellular replication. However, the molecular pathway associated with

the regulation of *phgdh* by Cdc42 is currently unknown. Cdc42 regulates diverse biological processes through its effects on the Jun amino-terminal kinase/stress-activated protein kinase and p38 mitogen-activated protein kinase pathways[50,51]. Whether Cdc42 regulates *phgdh* through one of these two known pathways or through an undiscovered pathway requires further study.

**Fig. 6 STM senses increased pyruvate and lactate levels in macrophages via CreBC, which activates SPI-2 gene expression through VrpB. a** Replication of the wt, *creB* mutant, and complemented STM strains in PMs. **b** Survival curves of mice infected i.p. with the wt, *creB* mutant, or complemented strain (*n* = 12 mice per group). **c** Liver and spleen bacterial burdens in mice infected with the wt, *creB* mutant, or complemented strain. **d** RT-qPCR analysis of *creB*, *creC*, *vrpB*, and *ssaG* mRNA levels in response to 1 mM pyruvate or lactate. **e** SPI-2 gene expression patterns in the wt and *creB* mutant strains profiled using RNA-seq (*n* = 3 independent experiments), with red representing higher and blue representing lower abundance. **f** EMSA of the *vrpB* promoter with purified CreB protein. **g** Fold enrichment of the *vrpB* promoter in CreB-ChIP samples. **h** EMSA of the *ssrA* and *ssrB* promoters with purified VrpB protein. **i** Fold enrichment of the *ssrA* and *ssrB* promoters in VrpB-ChIP samples. **j** Replication of the wt, *vrpB* mutant, and complemented strains in PMs. **k** Survival curves of mice infected i.p. with the wt, *vrpB* mutant, or complemented strain (*n* = 12 mice per group). **l** Liver and spleen bacterial burdens in mice infected with the wt, *vrpB* mutant, or complemented strain. **m** RT-qPCR analysis of *creB*, *vrpB*, and *ssaG* mRNA levels in intracellular STM. **n** RT-qPCR analysis of *ssaG* mRNA levels in the wt, *creB* mutant, or *vrpB* mutant strains in the presence or absence of 1 mM pyruvate or lactate. Data are presented as mean ± SD, *n* = 3 independent experiments (**a**, **d**, **g**, **i**, **j**, **m**, **n**), *n* = 7 mice per group (**c**, **l**). Images are representative of three independent experiments (**f**, **h**). *P* values were determined using two-tailed unpaired Student's *t* test (**a**, **j**), log-rank Mantel–Cox test (**b**, **k**), Mann–Whitney *U* test (**c**, **l**) or one-way ANOVA (**d**, **e**, **g**, **i**, **n**). *, **, *** *P* < 0.05, 0.01, 0.001, respectively (**e**). Source data are included in Source Data file.

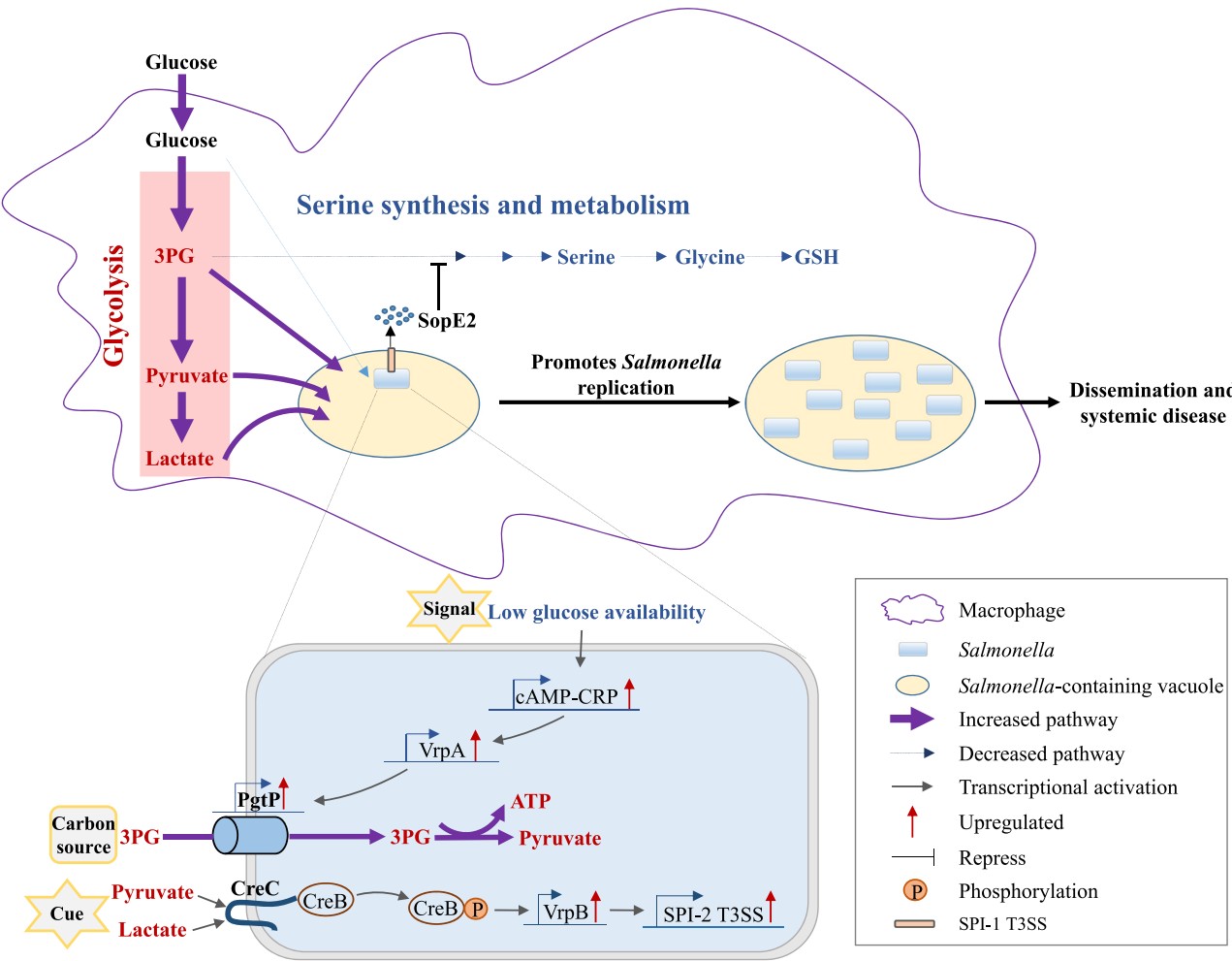

**Fig. 7 STM modulates macrophage glucose metabolism to support efficient intracellular replication and systemic infection.** STM translocates the effector SopE2 through the SPI-1 type III secretion system (T3SS), which represses macrophage serine synthesis and leads to the accumulation of glycolytic intermediates in macrophages. The decreased glucose level in macrophages induces the import of macrophage-derived 3PG as a main carbon source in order to support growth via VrpA in STM and senses increased pyruvate or lactate levels in macrophages to activate SPI-2 T3SS expression, thus promoting its intracellular replication and systemic infection.

The upregulation of SPI-2 genes in response to niche-specific cues in macrophages is critical for STM intracellular replication and systemic disease induction[52–55]. Indeed, inappropriate SPI-2 gene expression is energetically costly and can produce targets that might be recognized by the host immune system[56,57]. It has been shown that STM uses multiple regulatory pathways to control SPI-2 gene transcription. The main known regulators of those pathways include the positive regulators SsrA/B, PhoP/Q,

PhoB/R, OmpR/EnvZ, SlyA, PagR, and CRP, and the negative regulators PmrA/B and YdgT, which sense multiple environmental cues presented in macrophages, such as acidic pH, low magnesium concentrations, phosphate starvation, and the presence of antimicrobial peptides[53,54,58–62]. In this work, we found that STM also uses macrophage-accumulated pyruvate and lactate as cues to induce SPI-2 gene expression. Upon STM infection, the increased host glycolysis could continuously provide

pyruvate and lactate at a high rate, thus, pyruvate and lactate might represent some of the most reliable niche cues that STM can use to sustain SPI-2 gene expression. We also showed that macrophage pyruvate and lactate are sensed by the STM two-component system CreBC, which represents a new positive regulator of SPI-2 genes.

Host-derived pyruvate or lactate can be utilized by several other intracellular bacterial pathogens, such as the bacteria *Shigella* and *Brucella abortus*, as carbon sources for intracellular replication[63,64]. Although STM can utilize either pyruvate or lactate as a sole carbon source in vitro, its intracellular replication and pathogenicity were not influenced by the knockout of genes encoding pyruvate or lactate transporters, indicating that intracellular STM does not exploit host pyruvate and lactate as its main carbon sources. This might be because catabolism of pyruvate and lactate generates less energy than catabolism of other carbon substrates, such as glucose and 3PG, for intracellular STM. Thus, at least in macrophages, lactate and pyruvate act primarily as cues for STM.

When STM replicates within macrophages, macrophage glycolytic pathway is increased and STM uses the T3SS effector SopE2 to repress macrophage serine synthesis pathway, resulting in the accumulation of macrophage glycolytic intermediates, including 3PG, pyruvate, and lactate. The two-component system CreBC of STM senses the increased pyruvate and lactate levels of macrophages to activate the expression of SPI-2 genes via VrpB. The cAMP-CRP of STM in response to macrophage glucose limitation activates the expression of PgtP via VrpA. The increased expression of PgtP enables intracellular STM to utilize macrophage-derived 3PG as a major carbon source. It is known that cAMP-CRP also induces the transcription of SPI-2 genes[61]. Thus, it is possible that cAMP-CRP integrates the utilization of macrophage-derived 3PG with the enhancement of SPI-2 gene expression to achieve optimal intracellular replication of STM. However, it is worth noticing that a PhoP-activated small RNA PinT has been shown to repress *sopE2*, *crp*, as well as SPI-2 genes when STM is inside macrophages[65], suggesting the presence of negative regulatory mechanism to repress the over-expression of these virulence factors. How those positive and negative regulatory systems act in conjunction to achieve optimal STM intracellular replication is unclear and requires future studies.

Although STM strain 14028 s was used throughout this study, we also verified the main findings in SL1344, another commonly used mouse-virulent STM strain. SL1344 infection increased the levels of 3PG, pyruvate and lactate levels in macrophages (Supplementary Fig. 7a). Knockdown of Cdc42 decreased SL1344 intracellular replication (Supplementary Fig. 7b). Mutation of 3PG transporter gene in SL1344 decreased SL1344 intracellular replication and virulence (Supplementary Fig. 7c). Addition of lactate and pyruvate enhanced the transcription levels of *creB* and SPI-2 genes in SL1344 (Supplementary Fig. 7d). These results indicate that SL1344 induces the same metabolic changes in macrophages and responses to those changes in the same way as 14028 s does. Thus, other STM strains could use the same strategy to modulate and exploit the macrophage metabolic reprogramming for intracellular replication and virulence.

Taken together, our findings reveal that STM is able to reprogram macrophage glucose metabolism in order to initiate sufficient accumulation of macrophage-derived carbon sources and signals to support its intracellular replication and virulence. The accumulation of those glucose metabolic intermediates is the result of both host-driven and STM-driven macrophage metabolic reprogramming, which happen in parallel in the infected macrophages. STM reprograms the macrophage metabolism by using a T3SS effector to repress the host serine synthesis pathway

by repressing the key enzyme Phgdh, which could represent a promising target to control STM systemic infection.

## Methods

**Mice**. Male BALB/c mice (6–8 weeks old) obtained from Beijing Vital River Laboratory Animal Technology (Beijing, China) were used for all animal experiments in this work. Mice were housed under specific pathogen-free conditions with a 12 h light/dark cycle, at a temperature of $22 \pm 2\,°C$, and a relative humidity of $50 \pm 5\%$. Mice were fed a standard mouse chow diet. All animal studies were conducted according to protocols approved by the Institutional Animal Care Committee of Nankai University (Tianjin, China) and performed under protocol no. IACUC 2016030502.

**PM isolation and cell culture**. Mouse PMs were elicited using sodium periodate (Sigma #S1878) and isolated as follows. BALB/c mice were injected intraperitoneally (i.p.) with 3 mL of sterile 3% sodium periodate in PBS. On day 4, mice were killed, and PMs were collected by flushing the peritoneal cavity with 5 mL of ice-cold PBS. Peritoneal lavage fluid was centrifuged at $300 \times g$ for 5 min and resuspended in serum-free RPMI medium (Gibco #11879020). Cells were plated in 10-cm tissue culture plates at a density of $5 \times 10^6$ cells/plate or into 24-well tissue culture plates at a density of $2 \times 10^5$ cells/well. After 1 h, cells were washed twice with PBS to remove nonadherent cells and were cultured in RPMI medium supplemented with 10% (v/v) fetal bovine serum (FBS, Gibco #10100147). The medium was changed every 2 days, and PMs were used at 7 days post collection.

The RAW264.7 macrophage-like cell line (ATCC TIB-71) and HeLa cell line (ATCC CCL-2) were obtained from the Shanghai Institute of Biochemistry and Cell Biology of the Chinese Academy of Sciences (Shanghai, China) and cultured in RPMI medium supplemented with 10% (v/v) FBS (Gibco).

**Bacterial strains and growth conditions**. STM strain ATCC 14028 s was used as the wild-type (wt) strain throughout this study and for construction of the mutant and FLAG-tagged strains (Supplementary Table 4). STM mutant and FLAG-tagged strains were generated using the λ-Red recombinase gene replacement method[66]. To generate the *creBC-lux* reporter fusion, the amplification products of *creBC* promoter region was digested and cloned into the *XhoI–Bam*HI site of the plasmid pMS402, which carries a promoterless *luxCDABE* reporter gene cluster[67]. The primer sequences used for the generation and confirmation of strains are listed in Supplementary Table 5. Antibiotics, including chloramphenicol (Cm, 25 μg/mL), kanamycin (Km, 50 μg/mL), ampicillin (Ap, 100 μg/mL), and gentamicin (Gm, 10 or 100 μg/mL), were added when required for strain selection. Bacterial strains were routinely grown in Luria–Bertani (LB) medium (10 g/L tryptone, 5 g/L yeast extract, and 5 g/L NaCl) at 37 °C with shaking at 200 rpm or on LB agar plates.

For RT-qPCR or RNA-seq analysis, bacteria were initially cultured in LB medium, grown overnight at 37 °C, and then subcultured (1:100) in N-minimal medium (10 μM MgCl$_2$, 110 μM KH$_2$PO$_4$, 7.5 mM (NH$_4$)$_2$SO$_4$, 0.5 mM K$_2$SO$_4$, 5 mM KCl, 38 mM glycerol, 0.1% [w/v] casamino acids, and 80 mM MES [pH 5.5]) and grown for 6 h with shaking at 200 rpm. Subsequently, bacteria were collected for RNA isolation.

**In vivo mouse infection models**. Mice were injected i.p. with ~$10^4$ colony-forming units (CFUs) of the STM wt strain or with the indicated mutant or complemented STM strain. The mortality of infected mice was recorded daily and monitored for 20 days. For analysis of the bacterial burden in the liver and spleen of mice, infected mice were killed on day 3 post infection. Next, the liver and spleen were harvested, homogenized in ice-cold PBS, serially diluted, and plated on LB plates containing the appropriate antibiotics for CFU determination. For i.p. administration of D-(+)-glucose (Sigma #G7021), mice were injected i.p. with glucose (20 mg in 100 μl water) every other day starting 8 hpi in mouse infection models. For competitive infection, mice were injected i.p. with a 1:1 mix of two strains and killed on day 3 post infection to analysis of the bacterial burden in the liver and spleen. The CI value was calculated as (CFUs strain A recovered/CFUs strain B recovered)/(CFUs strain A inoculated/CFUs strain B inoculated).

**STM infection of macrophages**. PMs or RAW264.7 cells were infected with wt STM or the indicated mutant or complemented strains that grown to the late stationary phase at a multiplicity of infection (MOI) of $10^{66,67}$. Overnight-cultured bacterial strains (grown at 37 °C in LB medium with shaking for 18 h, late stationary phase) were pelleted by centrifugation, washed twice with PBS, resuspended and adjusted to an OD$_{600}$ of 0.6 (~$5 \times 10^8$ CFUs bacteria/mL) in RPMI medium. The bacterial culture was serially diluted to $4 \times 10^6$ CFUs/mL and opsonized in RPMI medium containing 10% normal mouse serum for 20 min, and were then added to macrophage monolayers (0.5 mL/well). Plates were centrifuged at $1000 \times g$ for 5 min to synchronize infection. After incubation for 30 min at 37 °C in 5% CO$_2$, the infected cells were washed three times with PBS and supplemented with fresh RPMI medium containing 100 μg/mL gentamicin for 2 h to remove extracellular bacteria. The monolayers were supplemented with RPMI medium containing 10 μg/mL gentamicin for the rest of the experiment. The infected cells were lysed with 1% Triton X-100 at the indicated time points after infection, and intracellular

bacteria were enumerated on LB agar plates for CFU analysis or collected for RT-qPCR analysis. The fold intracellular replication of the indicated STM strains was expressed as the CFUs recovered at 20 hpi relative to those at 2 hpi. To investigate the effect of macrophage glycolysis on intracellular STM replication, the glycolysis inhibitor PFK-15 (6 µM, Sigma #SML1009) was added at 1 hpi to avoid the possible effect of the inhibitor on STM entry into macrophages. To assess macrophage glucose uptake, PMs were infected with STM for 2, 8, and 20 hpi, washed three times with PBS, and incubated with 10 mM 2-NBDG (ENZO #PEP-23002) in glucose-free medium for 30 min. Changes in the intracellular fluorescence intensity of 2-NBDG were measured using a Spark multilabel plate reader (Tecan).

**STM infection of HeLa cells**. The HeLa cell infection assays were conducted with bacteria that were grown to log phase[68]. Overnight wt STM were diluted 1:100 into fresh LB medium and continued to grow at 37 °C with shaking to an $OD_{600}$ of 0.6 (log phase). Bacteria were pelleted, washed twice with PBS, resuspended in RPMI medium, serially diluted to $2 \times 10^6$ CFUs/mL, and then added to HeLa cells at a MOI of 10. Plates were centrifuged at $100 \times g$ for 5 min to synchronize infection. After incubation for 30 min at 37 °C in 5% $CO_2$, the infected cells were washed three times with PBS and supplemented with fresh RPMI medium containing 100 µg/mL gentamicin for 2 h to remove extracellular bacteria. The monolayers were supplemented with RPMI medium containing 10 µg/mL gentamicin for the rest of the experiment. The infected cells were lysed with 1% Triton X-100 at 2 and 6 hpi, and intracellular bacteria were enumerated on LB agar plates for CFU analysis. The fold intracellular replication was expressed as the CFUs recovered at 6 hpi relative to those at 2 hpi.

**Seahorse analysis**. The ECAR and OCR of PMs were determined using an XF24 extracellular flux analyzer (Seahorse Bioscience)[13,69]. In brief, PMs were plated in an XF24 plate at $2 \times 10^5$ cells/well, allowed to adhere for 1 h, washed twice with PBS, and cultured in RPMI medium supplemented with 10% (v/v) FBS. Cells were mock infected or infected with STM (MOI = 10) for 2, 8, and 20 h on the Seahorse plate as described above. Next, the culture medium was removed from the cells, and the cells were incubated in nonbuffered assay medium for 1 h at 37 °C in a measuring chamber without $CO_2$ input. For ECAR analysis, the assay medium was XF Base Medium (Seahorse Bioscience) supplemented with 1 mM L-glutamine. For OCR analysis, the assay medium was XF Base Medium supplemented with 10 mM glucose, 1 mM pyruvate, and 2 mM L-glutamine. The ECAR values were obtained by subtracting the nonglycolytic acidification values from the three averaged measurements after glucose injection. The basal OCR values were obtained by subtracting the nonmitochondrial respiration values from the values prior to oligomycin addition. ECAR and OCR values were normalized to the cell number.

**Metabolomics analysis**. PMs ($5 \times 10^6$) were seeded in 10-cm tissue culture plates and were mock infected or infected for 8 h with wt STM (MOI = 10). To avoid a time delay and associated changes in metabolite concentrations during the separation of bacterial contents from macrophages, we quenched the cell culture immediately after STM infection of 8 h and determined the combined metabolite concentrations from infected cells and STM. The cells were harvested and washed three times with ice-cold PBS to remove residual medium components. Subsequently, metabolites were extracted using a mixture of methanol/acetonitrile/dH₂O (40/40/20, v/v/v) in 0.1% formic acid. After incubation at −20 °C for 20 min, the samples were centrifuged at $12,000 \times g$ for 10 min at 4 °C, and the supernatant was collected for analysis of metabolite concentrations. Samples were analyzed with an ultra-high-performance liquid chromatograph (Acquity, Waters) coupled to a Q Exactive Hybrid Quadrupole-Orbitrap mass spectrometer (Thermo Fisher). Metabolites were separated on a Luna NH₂ 3 µm column ($2 \times 100$ mm) (Phenomenex). The mobile phases comprised 20 mM ammonium acetate adjusted to pH 9.0 with ammonium hydroxide (A) and acetonitrile containing 0.1% formic acid (B). The mass spectrometer with a heated electrospray ionization source was operated in positive/negative modes (ionization voltage, $+3.8$ kV/$-3.0$ kV; sheath gas pressure, 35.0 arbitrary units; capillary temperature, 320 °C), and was operated in full scan (70–1000 m/z, 70,000 resolution). Data were processed with Xcalibur 4.0 (Thermo Fisher), and ions were annotated according to accurate mass spectra by a search with Compound Discoverer 2.0 (Thermo Fisher). Metabolite levels were normalized to the cell number, and three biological replicates of each sample were analyzed.

To assess the contribution of STM metabolites to the combined metabolite concentrations, we cultured STM in RPMI medium and measured the concentrations of STM metabolites that were determined in the coculture assay. Each STM cell contained only 0.01–0.02% of the concentration of each corresponding combined metabolite measured in a single infected PM (Supplementary Table 1). Approximately 94% of the infected PMs contained no more than ten bacteria at 8 hpi (Supplementary Fig. 1e), confirming that the combined metabolites were dominated by the host metabolites.

**Serine and 3PG concentration measurements**. To assess the changes in the intracellular concentrations of serine and 3PG during STM infection, PMs cultured in serine-free RPMI medium (Boster Bio, Wuhan, China) were infected with wt STM or heat-inactivated STM for 8 h (MOI = 10). The cells were washed three

times with ice-cold PBS and were then lysed with dH₂O for 30 min on ice. The samples were then centrifuged at $12,000 \times g$ for 10 min at 4 °C, and the supernatants were collected for measurement of serine and 3PG concentrations. Serine concentration was measured using a Mouse Ser ELISA Kit (Jianglai Bio, Shanghai, China) according to the manufacture's manual. 3PG concentration was measured using a U-3000 UPLC system (Thermo Fisher) with a Bio-rad Aminex HPX-87H column and a RID detector. The mobile phase was 5 mmol/L sulfuric acid solution at a flow rate of 0.6 mL/min at 60 °C. Standard solutions of serine were prepared in 5 mM $H_2SO_4$.

**SiRNA and overexpression plasmid transduction**. For Phgdh or Pfkfb3 siRNA transduction, RAW264.7 cells were transfected with Phgdh siRNA (si-Phgdh) or Pfkfb3 siRNA (si-Pfkfb3), or negative control siRNA (NC, si-Control) using INTERFERin (Polyplus Transfection #PT-409-10) 48 h before the macrophage infection assay. For Cdc42 siRNA transduction, HeLa cells were transfected with cdc42 siRNA (si-Cdc42) or NC siRNA using INTERFERin 48 h before the HeLa cell infection assay. The siRNA and NC siRNA constructs were synthesized by Guangzhou Ruibo Biotechnology Co., Ltd. (Guangzhou, China). The sense sequences of the Phgdh siRNA, Pfkfb3 siRNA, and Cdc42 siRNA were 5′–GGAGC GGGAGATCGAGAAC–3′, 5′–GTGAGGAACTAACCTATGA–3′, and 5′–CCAT CGGAATATGTACCAA–3′, respectively. Phgdh expression, Pfkfb3 expression, and Cdc42 expression were then detected by western blotting 48 h after transfection.

For Phgdh-overexpressing plasmid transduction, Phgdh cDNA was PCR amplified with a PrimeScript RT Reagent Kit with gDNA Eraser (TaKaRa #RR047A) from cDNA synthesized from RAW264.7 cell RNA. Full-length phgdh was digested with the BamHI and XbaI restriction enzymes and was then ligated into the pcDNA3.1 (+) vector (Invitrogen #V87020). Plasmids were cotransfected into RAW264.7 cells (60–80% confluency) using jetPRIME (Polyplus Transfection #PT-114-07) for 48 h. The following primers were used for phgdh overexpression: 5′–CGCGGATCCCATGGCCTTCGCAAATCTGCGCA–3′ and 5′–TGCTCTAGAT CAGAAGCAGAACTGGAAAGCCTCC–3′.

**Lactate and pyruvate concentration measurements**. To assess the changes in lactate levels of PMs during STM infection, PMs were mock infected or infected with wt STM for 2, 8, and 20 h. The supernatants were collected for measurement of lactate concentration. To assess the changes in lactate and pyruvate levels of RAW264.7 cells upon Pfkfb3 knockdown, RAW264.7 cells were transfected with Pfkfb3 siRNA or negative control siRNA for 48 h, and the supernatants were collected for measurement of lactate concentration. The RAW264.7 cells were washed three times with ice-cold PBS and were then harvested for measurement of pyruvate concentration. Lactate concentration was measured by using a Lactate assay kit (Abcam #ab65330) according to the manufacture's manual. Pyruvate concentration was measured by using a Pyruvate assay kit (Abcam #ab65342) according to the manufacture's manual. Results were normalized to the cell number.

**Western blotting**. Protein samples from cultured RAW264.7 cells were prepared by lysis with radioimmunoprecipitation assay buffer (Solarbio #R0010) followed by centrifugation at $15,000 \times g$ at 4 °C for 5 min. The supernatants were collected, heated at 95 °C for 10 min and used for sodium dodecyl sulfate-polyacrylamide gel electrophoresis. Protein samples were separated on a SurePAGE™ gel (GenScript #M00653) and then transferred to polyvinylidene difluoride membranes using a wet transfer system. Membranes were blocked with tris-buffered saline with 0.1% (w/v) Tween 20 (TBST) containing 5% skim milk for 2 h at room temperature and were then incubated with the primary antibody followed by the horseradish peroxidase-conjugated goat anti-rabbit IgG secondary antibody (1:5,000 dilution, Sparkjade #EF0002). The primary antibody used including: anti-PHGDH antibody (1:1000 dilution, Cell Signaling Technology #13428), anti-PFKFB3 antibody (1:1000 dilution, Abcam #ab181861), anti-CDC42 antibody (1:1000 dilution, Abcam #ab187643), and anti-Actin antibody (1:2000–1:5000 dilution, Abcam #ab179467). Immunoreactions were detected using SuperSignal™ West Pico PLUS Chemiluminescent Substrate (Genestar #E170-01). Images were acquired using an Amersham™ Imager 600 System (General Electric Company). Western blotting bands were quantified using Image J180 software.

**Confocal immunofluorescence microscopy**. To visualize STM infection of PMs, PMs were seeded on glass coverslips and infected as described above. At 2 and 20 hpi, cells were fixed with 3% paraformaldehyde for 15 min and washed extensively with PBS. Cells were then permeabilized with 1% Triton X-100 for 20 min and blocked with 5% bovine serum albumin in PBS for 30 min. Subsequently, cells were stained with a mouse anti-Salmonella typhimurium LPS antibody (1:100 dilution, Abcam #ab8274) for 1 h and the secondary antibody goat anti-mouse IgG (FITC) (1:200 dilution, Abcam #ab6785) for 1 h. Cells were then incubated with DAPI (Invitrogen #D21490) for 2 min. Cell images were acquired using a confocal laser scanning microscope (Zeiss LSM800) and analyzed with ZEN 2.3 (blue edition).

**RNA isolation**. Bacterial and macrophage RNA were extracted using TRIzol reagent (Invitrogen #15596018) according to the manufacturer's protocol. RNA

samples were purified with an RNeasy Mini Kit (Qiagen #74104), including an on-column DNase I (Qiagen #79254) digestion step. RNA concentrations were measured using a NanoDrop 2000 spectrophotometer (NanoDrop Technologies).

**RNA sequencing and analyses**. RNA-seq was conducted using PMs and STM. PMs used for RNA-seq were cultured in RPMI medium supplemented with 10% (v/v) FBS and were mock infected or infected with STM (MOI = 10) for 8 h. For bacterial RNA-seq, RNA was extracted from wt or *creB* mutant strains grown in N-minimal medium (OD$_{600}$~0.6). cDNA libraries were generated using an NEBNext® Ultra™ Directional RNA Library Prep Kit for Illumina® (New England Biolabs) following the manufacturer's recommendation, and sequencing was performed by Majorbio Bio-Pharm Technology Co., Ltd. (Shanghai, China). RNA-seq was performed on triplicate samples. Sequence reads were mapped to the mouse genome (*Mus musculus*, GCF_000001635.26) or the STM ATCC 14028 s reference genome (CP001363 and CP001362) using Bowtie (version 2.2.3). The read numbers mapped to each gene were calculated using HTSeq (version 0.6.1). The fragments per kilobase of transcript per million mapped reads method was used to evaluate the expression values of each gene. Differentially expressed genes in treated cells compared to in untreated control cells were identified using DESeq2 (|log$_2$ fold change| > 1 and $P < 0.05$). $P$ values were adjusted using the Benjamini and Hochberg approach to control the false discovery rate.

**Real-time quantitative PCR**. RT-qPCR was performed in a QuantStudio 5 Real-Time PCR system (Applied Biosystems) using Power SYBR Green PCR Master Mix (Applied Biosystems #4367659) following the manufacturer's protocols. cDNA synthesis was performed by using a PrimeScript RT Reagent Kit (TaKaRa #RR036A). Each RT-qPCR was performed in triplicate in a total reaction volume of 20 μL containing 10 μL of Universal SYBR Green Master mix, 1 μL of cDNA, and two gene-specific primers. The expression levels of STM transcripts were normalized to that of the 16 S *rRNA* gene, whereas those of macrophage transcripts were normalized to β-actin expression. The fold change in the expression of each target gene was estimated using the 2$^{-\Delta\Delta Ct}$ method.

**Bioluminescent reporter assays**. To determine intracellular *creBC* expression using the *lux* bioluminescent reporter, PMs were infected with the STM wt carrying *creBC-lux* fusion as described for the macrophage infection assays. At indicated time points (2, 8, and 20 hpi), the cells were lysed with 1% Triton X-100. A 200 μl sample of the cell lysates was loaded into a 96-well white assay plate (Corning) to measure luminescence using the Spark multimode microplate reader (Tecan). A 100 μl sample of the cell lysates were serially diluted and plated on LB agar plates to enumerate intracellular bacterial CFUs. Luminescence values were normalized to intracellular bacterial CFUs to determine the expression level of CreBC.

**Electrophoretic mobility shift assay**. PCR fragments encompassing the promoter regions of the indicated genes were amplified using genomic DNA of the STM wt strain as a template. Purified DNA (40 ng) was mixed with various amounts of purified proteins in 20 μL of gel binding buffer (10 mM Tris-HCl [pH 7.4], 100 mM NaCl, 1 mM EDTA, 10% v/v glycerol, 0.1 mM DTT, and 10 μg/mL bovine serum albumin) containing 2 mmol of cAMP (only for CRP protein, Sigma). After incubation at 37 °C for 30 min, the samples were separated by 6% PAGE at 100 V for 50 min. Gels were stained with GelRed for 10 min and imaged using a gel imaging system (Tanon).

**Chromatin immunoprecipitation**. FLAG-tagged STM strains were grown overnight in LB medium and then diluted 1:100 into N-minimal medium and continued to grow at 37 °C with shaking for 6 h. Bacterial cells were treated with 1% formaldehyde for 25 min at room temperature. The cross-linking reaction was quenched by the addition of 0.5 M glycine. Bacterial pellets were washed twice with PBS and were then sonicated to generate DNA fragments of ~500 bp. Cell debris was pelleted by centrifugation at 4 °C, and the supernatant was used as the cell extract for the immunoprecipitation (IP) experiment. Protein-DNA complexes were incubated with 50 μL of an anti-FLAG mouse monoclonal antibody (1:1,000 dilution, Sigma #F1804) for 8 h and were then enriched with Protein A magnetic beads (Invitrogen #88846). IP for the mock ChIP sample (negative control) was performed using different aliquots without the addition of antibodies.

**Chromatin IP sequencing library construction and sequencing**. ChIP sample DNA and mock ChIP sample DNA ends were blunted, and the DNA was A-tailed and ligated to Illumina adapters. Following ligation, size selection was performed through gel electrophoresis via excision of DNA fragments between 150 and 250 bp. After gel purification, 18 cycles of PCR amplification were performed using the Illumina genomic DNA primers 1.1 and 2.1. Sequencing libraries were prepared using a NEXTflex™ Rapid DNA-Seq Kit (Bioo Scientific). High-throughput sequencing was carried out on the Illumina HiSeq 2500 platform. ChIP-seq was performed once for the ChIP sample and the mock ChIP sample. Raw reads were first processed with Cutadapt (v1.9.1) to remove adapters and low-quality sequence (quality scores < 20). The trimmed reads were mapped to the STM ATCC 14028 s

genome (CP001363 and CP001362) using BWA mem (version 0.7.12). Only uniquely mapped reads with an alignment score ≥20 were used for peak calling. After mapping reads to the reference genome, the MACS2 (v2.1.0) peak finding algorithm was used to identify regions of IP enrichment over control. A $q$ value threshold of enrichment of 0.05 was used for all data sets.

**ChIP-qPCR**. ChIP experiments were performed as described above. Quantification of *vrpA*-, *creB*-, and *crp*-bound DNA was carried out by RT-qPCR using Power SYBR Green PCR Master Mix in an ABI QuantStudio 5 Real-Time PCR system. The relative enrichment of DNA regions of interest was calculated using the 2$^{-\Delta\Delta Ct}$ method.

**Measurement of bacterial cAMP concentration**. RAW264.7 cells were infected with the STM wt strain as described above. At 2, 4, 8, and 20 hpi, the infected cells were lysed with 1% Triton X-100, and intracellular bacteria were pelleted by centrifugation. The cAMP concentration in intracellular bacteria was measured by a Direct cAMP ELISA Kit (Abcam #ab133051) following the manufacturer protocol. Cell lysates were also plated, and cAMP concentrations were normalized to intracellular bacterial CFUs.

**Statistical analysis**. Statistical significance was analyzed with GraphPad Prism 8.0.1 software (GraphPad Inc., San Diego, CA) using the two-tailed unpaired Student's $t$ test, one-way analysis of variance, log-rank (Mantel–Cox) test, or Mann–Whitney $U$ test according to the test requirements (as stated in the figure legends). A $P$ value of < 0.05 was considered statistically significant.

**Reporting summary**. Further information on research design is available in the Nature Research Reporting Summary linked to this article.

## Data availability
The RNA-seq and ChIP-seq data acquired in this study are available in the NCBI Sequence Read Archive (SRA, PRJNA561041). The metabolomic data have been deposited in MetaboLights (MTBLS2347). Source data are provided with this paper.

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

## Acknowledgements

This work was funded by the National Key Programs for Infectious Diseases of China, grant no. 2017ZX10303405 (to L.W.); the National Natural Science Foundation of China, grant no. 31820103002 (to L.W.), 31530083 (to L.W.), 31770144 (to L.W.), 32070133 (to L.F.), 81871624 (to L.F.), 31800126 (to L.J.), 31970084 (to D.H.), and 31970179 (to Z.C.); and the National Key R&D Program of China, grant no. 2018YFA0901000 (to L.F.).

## Author contributions

L.W. and L.F. designed the research; L.J., P.W., X.S., H.Z., S.M., J.W., R.L., S.M., W.L., J.Y., and X.L. performed the research; Z.C., C.Y., and D.H. provided technical support and insights; H.Z. and C.Y. assisted with the metabolomic study; L.J., P.W., X.S., and H.Z. analyzed the data; and L.W., L.F., and L.J. wrote the manuscript.

## Competing interests

The authors declare no competing interests.
