## [Peer Review File · Nature Communications]

REVIEWER COMMENTS

Reviewer #1 (Remarks to the Author):

In this manuscript Jiang et al., describe that *Salmonella Typhimurium* 14028s strain (STM) reprograms glycolysis in peritoneal and RAW macrophages. They have observed that STM increases glucose uptake, glycolytic intermediates and some of the TCA cycle derived metabolites such as succinate and fumarate. Importantly, they have found increased accumulation of phosphoglycerates which are precursors for Serine synthesis. Authors have also found that the SPI-I effector protein SopE2 prevents Serine biosynthesis in macrophages by repressing the expression of *phgdh* although how this gene in particular is transcriptionally repressed is unknown. They have identified *pgtP* as the gene that is responsible for the import of 3PG by the bacteria but *pgtP* is repressed by glucose. The authors have also identified *VrpA* as a potential regulator of *pgtP*. Furthermore, they have also identified the two-component system *CreBC* senses pyruvate and lactate which in turn activates SPI-2 genes. Also, they have deciphered that *CreB* regulates SPI-2 gene expression by binding to the promoter region of *VrpB*. The study is interesting because the authors have done extensive work to identify the link between metabolic changes and the expression of SPI-2 genes which is highly critical for *Salmonella* pathogenesis. However, this does not completely explain the metabolic changes occurring in macrophages upon infection. The authors are right in claiming that glycolysis is reprogrammed but the data does not convince to show that increased glycolysis is the cause for increased activation of *CreB* 2 component system dependent SPI-2 genes. The study would highly benefit if the authors have focus only on the 2-component system they have newly identified which senses metabolic changes rather than elaborating on the metabolic changes in macrophages which is less convincing especially the role of increased glycolysis in pathogen clearance.

Overall the link between the various virulence factors identified and the metabolic changes observed are less convincing.

Please see below for more specific comments.

The title is *Salmonella* promotes its intracellular replication by reprogramming macrophage glucose metabolism..... which is very confusing. It speaks of reprogramming glucose metabolism to increase glycolysis and inhibit serine synthesis. The focus of the paper has to be rather on the identification of the virulence factors that sense metabolic changes.

The authors have used *Salmonella enterica* serovar *Typhimurium* in their experiments which is one of the many serovars. Therefore, it is important to specify *Salmonella Typhimurium* in the title. *Salmonella* alone is misleading.

It is also not clear to me why the authors address *Salmonella* as a model pathogen in the abstract when the whole manuscript is based on identifying virulence factors of *Salmonella*. In some places it is mentioned as *Salmonella* and in some as STM. It needs to be consistent.

What is the rationale for using peritoneal macrophages?

In the infection protocol they mention that they pellet an overnight culture and use it to infect the macrophage monolayer. How do the authors know it is MOI 10? The infection protocol doesn't seem to be right. This is the basis for all the data.

Moreover, the infection protocol for HeLa cells are very different from that of macrophages. *Salmonella Typhimurium* are well known inducers of various cell death pathways. How do the authors correlate the metabolic data to cell death?

Peritoneal macrophages will be highly activated while isolating them due to the injection of sodium periodate. It is surprising that peritoneal macrophages withstood *Salmonella* infection for 20h and the authors find glycolysis to be high.

Glucose uptake peaks at 8h pi but decreases by 20h – this observation has not been stated.

How do the authors explain for the increase in succinate, fumarate and α -ketoglutarate although the transcriptomics data show that the respective enzymes such as *SdhA* are decreased? Increase in succinate also means that flux through TCA occurs if so, how does pyruvate accumulate?

The authors will have to present the original sea horse graph and data.

For the next part of the data to show the effect of glycolysis authors have switched to RAW macrophages which is not the same as peritoneal macrophages and it is also not clear as why *Pfkfb3* was silenced.

Bacterial burden is represented as fold change which is not appropriate. For instance, 0.2-fold

change upon over-expressing Phgdh is not significant. Data needs to be represented as a log₁₀ CFU as has been shown for in vivo data.

How does SopE2 transcriptionally repress phgdh?

The authors claim PgtP as the transporter of G3P, pyruvate and lactate and they refer to an article from 1975. However, in the cited paper it is mentioned as PgtA. Is PgtP and PgtA synonymous.

Data shows that reduced glucose in RPMI increases the expression of PgtP. When authors performed metabolite analysis macrophages should have been grown in RPMI which contains 2g/L under such conditions PgtP expression is repressed. It is then contradictory that PgtP is the virulence factor that enables the bacteria to survive in macrophages by utilising G3P.

Is the macrophage metabolism affected when macrophages are infected with the pgtP mutant?

There is no substantial evidence that supports the claim that reprogramming glycolysis increases glycolysis.

Is the expression of pgtP altered when pfkfb3 is silenced?

It would be convincing if the authors show the expression of the CreB 2 component system using reporter plasmids in macrophages.

10⁴ bacteria have been injected i.p for in vivo experiments. To my knowledge it is too high for the animals to survive even 6-7 days.

Finally, Salmonella Typhimurium 14028s has been reported as a less invasive strain as they lack SPI-1 virulence factors such as SopE. How does this compare to a more invasive strain such as SL1344?

Reviewer #2 (Remarks to the Author):

In this study, Jiang, Wang, Song, et al investigate how the enteric pathogen Salmonella Typhimurium manipulates macrophage metabolism to support intracellular replication. Consistent with previous results, the authors provide evidence that in peritoneal macrophages, Salmonella induces an upregulation of flux through glycolysis, inhibition of serine synthesis, and decreased activity in the TCA cycle. Manipulation of macrophages with siRNA and inhibitors recapitulated these effects in the absence of Salmonella infection, and enhanced Salmonella replication. A mutant unable to transport 3-phosphoglycerate (pgtP) is attenuated for macrophage replication in vitro and in a mouse infection model. The authors identify the type three secretion system effector SopE2 as the cause for Salmonella-induced changes in serine biosynthesis. Furthermore, the authors uncover evidence suggesting that there are two novel Salmonella signaling pathways: low glucose levels result in CRP/cAMP-dependent transcriptional activation of vrpA, which in turn enhances transcription of pgtP. Lactate and pyruvate are sensed by the CreCB system, which promotes T3SS-2 transcription via VrpB. The authors conclude that inside macrophages, Salmonella inhibits serine synthesis to acquire the accumulated metabolite pre-cursors.

The topic of metabolism in the context of host-microbe interactions is timely and should be of great interest. There are several original findings and the concept that a type three secretion effectors alters host metabolism, which in turn facilitates pathogen growth, is intriguing. The writing is clear and the data presented in a logical manner. I have two major concerns regarding the validity of the overall conclusions:

1. The importance of SopE2 inhibiting serine synthesis in the mouse model is unclear. If the author's model is correct, one would predict that the SopE2 mutant is as attenuated as the other mutants tested in this study (e.g. PgtP). Since this is a central tenet of this work, the mutant is available, and the mouse model is established, this key experiment should be performed.
2. I have significant concerns regarding the interpretation of the experiments involving regulatory mutants, i.e. the vrpA and vrpB mutant, in the mouse model. The authors provide evidence that VrpA regulates PgtP transcription and VrpB influences SPI2 transcription. However, it is very common that regulatory proteins affect more than one target. For example, the vrpB mutant could be attenuated (Fig. 6k and I) because SPI2 or other (metabolic) genes are poorly expressed in general and not because SopE2 expression is affected. Similarly, the vrpA mutant could be attenuated through action of another, unknown pathway. One simple approach to delineate cause-

end-effect relationships and exclude these pleiotropic effects is through epistasis experiments. For example, one would expect that introducing a *vrpA* mutation into a *pgtP* mutant background (*vrpA pgtP* double mutant) should not further attenuate the strain. In contrast, if the *vrpA pgtP* double mutant was more attenuated than the *vrpA* mutant or the *pgtP* mutant, this would indicate off-target, pleiotropic effects or inconsistencies in the model. As such, the fold replication and the virulence in a mouse model for the following double mutants must be determined: *vrpA pgtP*, *creCB vrpB*, *vrpB SPI2*, and *PgtP SopE2*. In the absence of these data, several major conclusions are insufficiently supported.

Specific Comments:

3. Macrophage reprogramming and utilization of glucose by *Salmonella* has been reported before by Eisele et al. (PMID: 23954156). Their findings align very well with the findings in this report and therefore, it is unclear to me why the Eisele paper is not cited. It certainly should be.
4. Line 187: Should read "RT-qPCR" instead of "qPCR" and "transcription" instead of "expression"?
5. Line 240: The title of this section is a bit of an overstatement. This is only based on the analysis of two types of mutants and one time point. It is conceivable that other, unknown transporters of glucose exist and were not considered. Also, one has to consider the possibility that changing metabolism creates pleiotropic effects, such as the accumulation of toxic metabolites, which can affect fitness. My recommendation is to rephrase that section title.
6. Line 356 and Fig 7: Should read "cue" and not "signal". It is unlikely that the mammalian host has evolved to produce pyruvate and lactate to communicate with bacteria. There are likely other instances in the manuscript where signal should be replaced with cue.
7. The authors should consider rewording the title. Most of the manuscript is devoted to *SopE2*-mediated reprogramming of metabolism and coordinating bacterial gene expression.

Reviewer #3 (Remarks to the Author):

In this manuscript, Jiang et al. describe interesting findings showing that *Salmonella* Typhimurium (STM) modify glucose metabolism of macrophage to promote its intracellular replication and thus its virulence. Specifically, the authors demonstrate that STM upregulates the expression of enzymes for glycolysis and decreases the expression of enzymes for serine synthesis, which leads to the accumulation within macrophages of glycolytic intermediates, such as 3-phosphoglycerate (3PG), pyruvate and lactate. The mechanism by which STM represses serine synthesis, as well as a mechanism for the uptake of macrophage 3PG to be used by STM as a carbon source, which involves sensing of low glucose, were defined. The authors also demonstrate that STM senses pyruvate and lactate to induce expression of the SPI-2 genes, which are known to be essential for STM intracellular replication, and that the regulatory cascade responding to pyruvate and lactate is required for STM virulence.

I find this study as technically sound and relevant, with novel results, as well as the manuscript well written. The findings from this study further expand the knowledge about the different mechanisms used by pathogenic bacteria to manipulate hosts; in this case, by reprogramming metabolism.

However, I recommend to consider findings from some other reports for the integration of a more complete picture about the mechanisms used by *Salmonella* to survive/replicate inside macrophages. Additionally, the role of the pyruvate/lactate-CreB-VrpB-SPI2 regulatory cascade in *Salmonella* intracellular replication should be further assessed. See comments below.

Comments

-Whereas this study shows that STM infection enhances glycolysis in macrophages, the report by Sanman et al. (2016) indicates that STM infection reduces glycolytic flux also in macrophages. Please discuss about this discrepancy.

-How is increased the expression of PMs glycolytic genes? Is SopE also involved in their upregulation?.

-Are there previous studies involving Cdc42 on gene expression or specifically on regulation of genes for metabolism, such as those for serine synthesis?.

-It is unclear how VrpA was identified as a regulator of *pgtP* by using BPROM, a tool for prediction of bacterial promoters. Same for CRP as a regulator of *vrpA*. Please describe with more detail this procedure.

-What kind of regulator is VrpA?. Is it controlled by any of the regulators involved in the expression of genes for STM intracellular replication?. You could easily check this in SalComRegulon.

-Is CRP required for intracellular replication of STM?. Discuss about what is known on the role of CRP in *Salmonella* virulence.

-Consider for Discussion previous findings reported by Westermann et al. (2016; doi: 10.1038/nature16547), showing the effect of STM infection on macrophage transcriptome, including data indicating repression of SopE2, SPI-2 and CRP by PhoP-PinT when *Salmonella* is intracellularly. How these findings and the results from this study can converge to explain all the mechanisms used by STM for intracellular replication?.

-While it is clear that the regulatory pathway CreB-VrpB induces expression of the SPI-2 genes in response to pyruvate or lactate, and that it is required for STM intracellular replication, the STM mutants blocked in the uptake of pyruvate or lactate are not affected in intracellular replication, which could be due to the fact that pyruvate and lactate probably activate by separate the CreB-VrpB regulatory cascade. To further investigate the role of the pyruvate/lactate-CreB-VrpB-SPI2 regulatory cascade in *Salmonella* virulence, you can test a STM double mutant, blocked in the uptake of both pyruvate and lactate, for intracellular replication and for activation of SPI-2 (*ssaG*) expression in response to pyruvate and lactate. This could make stronger your conclusion indicating that the SPI-2 genes are induced inside macrophages in response to pyruvate or lactate for intracellular replication and thus for virulence of *Salmonella*. Is there any domain in CreC that could be involved in sensing pyruvate and/or lactate?.

-Integrate in your model the findings from this work and those from other studies. For instance, this work show that low glucose as well as pyruvate and lactate are cues detected by *Salmonella* inside macrophages through mechanisms involving the regulators CRP, VrpA, CreC/CreB and VrpB, being the CreC/B-VrP pathway necessary for induction of the SPI-2 genes; SPI-2 is essential for *Salmonella* replication within host cells. However, several other studies have demonstrated that *Salmonella* also senses other signals present inside macrophages, such as acidic pH and low phosphate and magnesium, through regulators well characterized like PhoP/Q, OmpR/EnvZ and SsrA/B, which induce expression of the SPI-2 genes inside host cells.

-I hope that the authors will find my recommendations reasonable and useful.

Víctor Bustamante

Responses to reviewers

We thank the reviewers for their careful review. We have heeded their suggestions and have performed additional experiments to clarify and strengthen the findings. We believe, by doing so, we have improved and strengthened the manuscript substantially and we thank the reviewers for this. Our point-by-point responses to their specific comments are provided below.

Reviewer #1 (Remarks to the Author):

In this manuscript Jiang et al., describe that Salmonella Typhimurium 14028s strain (STM) reprograms glycolysis in peritoneal and RAW macrophages. They have observed that STM increases glucose uptake, glycolytic intermediates and some of the TCA cycle derived metabolites such as succinate and fumarate. Importantly, they have found increased accumulation of phosphoglycerates which are precursors for Serine synthesis. Authors have also found that the SPI-I effector protein SopE2 prevents Serine biosynthesis in macrophages by repressing the expression of phgdh although how this gene in particular is transcriptionally repressed is unknown. They have identified pgtP as the gene that is responsible for the import of 3PG by the bacteria but pgtP is repressed by glucose. The authors have also identified VrpA as a potential regulator of pgtP. Furthermore, they have also identified the two-component system CreBC senses pyruvate and lactate which in turn activates SPI-2 genes. Also, they have deciphered that CreB regulates SPI-2 gene expression by binding to the promoter region of VrpB.

The study is interesting because the authors have done extensive work to identify the link between metabolic changes and the expression of SPI-2 genes which is highly critical for Salmonella pathogenesis. However, this does not completely explain the metabolic changes occurring in macrophages upon infection.

Response:

Thank you for your patient and thoughtful reading as well as the constructive comments and advices about our manuscript. We have revised the manuscript based on your comments and suggestions.

You are right that this work does not completely explain the metabolic changes occurring in macrophages upon infection by *Salmonella enterica* serovar Typhimurium (STM), as we focused on some of the changes which may influence the replication of STM. Our investigations showed that STM infection increased macrophage glycolysis and STM used the SPI-1 effector SopE2 to repress macrophage serine synthesis, triggering glycolytic intermediate accumulation in macrophages. We also showed that SopE2 can repress the expression of 3-phosphoglycerate dehydrogenase (Phgdh), the key enzyme in the serine synthesis pathway.

The authors are right in claiming that glycolysis is reprogrammed but the data does not convince to show that increased glycolysis is the cause for increased activation of CreB 2 component system dependent SPI-2 genes.

Response:

In the original manuscript, we showed that STM infection increased and also reprogrammed macrophage glycolysis. We found that STM can sense the accumulated macrophage pyruvate and lactate *via* the CreB/C two-component system to activate SPI-2 gene expression. As the accumulation of lactate is a key feature of the increase in macrophage glycolysis (*FEBS J*, 2018, 285:2146–60), we concluded that the increased macrophage glycolysis activates STM SPI-2 gene expression *via* CreBC.

The Reviewer is right that we did not show directly that the increased glycolysis is the cause for increased activation of CreBC dependent SPI-2 genes. We thank the Reviewer for pointing out this.

We have now performed two additional experiments to further confirm that the increased macrophage glycolysis activates STM SPI-2 gene expression *via* CreBC:

i) According to this comment and the No. 19 comment, we have now investigated whether the inhibition of macrophage glycolysis influences the intracellular expression levels

of *creB*, *vrpB* and *ssaG* (a representative SPI-2 gene) in wt and $\Delta creB$ strains. We blocked macrophage glycolysis via siRNA knockdown of Pfkfb3, the rate-limiting enzyme of glycolytic metabolism. We confirmed the decrease in lactate and pyruvate production upon Pfkfb3 knockdown (Supplementary Fig. 2b of the revised manuscript). Through RT-qPCR analysis, we found that wt strain can respond to the decreased macrophage glycolysis upon Pfkfb3 knockdown, as indicated by the decreased intracellular *creB*, *vrpB* and *ssaG* transcription (Supplementary Fig. 6c of the revised manuscript). These results suggest that CreBC responds to the increased macrophage glycolysis to activate the downstream *vrpB* and SPI-2 genes. These results have now been added to the revised manuscript (lines 174–176, 417–420, 477–479 and Supplementary Figs. 2b, 6c).

ii) According to this comment and the No. 8 comment of reviewer #3, we generated the *lldp cstA-ybdD* double mutant of STM which cannot uptake either pyruvate or lactate. Through RT-qPCR analysis, we found that *lldp* and *cstA-ybdD* double mutation did not influence *ssaG* transcription in the presence of pyruvate and lactate in N-minimal medium (Supplementary Fig. 6g of the revised manuscript), revealing that the utilization of pyruvate and lactate is not related with SPI-2 gene expression. The result further indicates that pyruvate and lactate act as cues to induce SPI-2 expression. This information has been added to the revised manuscript (lines 430–433 and Supplementary Fig. 6g).

The study would highly benefit if the authors have focus only on the 2-component system they have newly identified which senses metabolic changes rather than elaborating on the metabolic changes in macrophages which is less convincing especially the role of increased glycolysis in pathogen clearance. Overall the link between the various virulence factors identified and the metabolic changes observed are less convincing.

Response:

Thank you for the suggestions.

We agree that more attention should be paid on the CreBC two-component system which promotes STM pathogenesis through sensing the metabolic changes in macrophages.

According to this suggestion and the No. 8 comment of reviewer #3, we have now performed additional experiments to further confirm that the increased pyruvate and lactate in macrophages upon STM infection activates STM SPI-2 gene expression *via* the CreBC two-component system, please see the previous response.

You are right that the increase in macrophage glycolysis is beneficial to the clearance of pathogens by macrophages, since glycolysis not only boosts inflammatory responses and cytokine production to control pathogen replication, but also supports the generation of reactive oxygen and nitrogen species, itaconate, and prostaglandins, all of which contribute to macrophage antimicrobial activities. While macrophages increase their glycolytic metabolism upon activation by microbial ligands, several intracellular bacterial pathogens have been shown to reprogram the host metabolism in a targeted, specific manner that benefits the pathogen for intracellular replication (eg. *Legionella pneumophila* and *Francisella tularensis*; *Cell Host Microbe*, 2017, 22:302–16; *J Immunol*, 2016, 196:4227–36). In this work, we found that STM can also reprogram macrophage metabolism to accumulate glycolytic intermediates, which are utilized by STM as carbon sources and act as cues to induce STM virulence gene expression, for optimal intracellular replication.

We agree that the link between the various virulence factors identified (CreBC–VrpB–SPI-2) was less convincing in the original manuscript. We demonstrated, in the original manuscript, the positive regulations of VrpB by CreB and SPI-2 by VrpB *via* mutation, RT-qPCR, EMSA and ChIP-qPCR analysis (Fig. 5f–i; Extended Data Fig. 4f,g of the original manuscript). However, whether the contribution of CreB to STM virulence is entirely dependent on VrpB, and whether the contribution of VrpB to STM virulence is entirely dependent on SPI-2, were not clear.

According to this suggestion and the No. 2 comment of reviewer #2, we have now performed additional experiments to further address the regulatory link between the CreBC two-component system and SPI-2 genes. We used two double mutant strains, $\Delta creB\Delta vrpB$ and $\Delta vrpB\Delta ssrB$, to do so.

Regarding the regulatory link between CreB and VrpB, we compared the virulence between $\Delta creB\Delta vrpB$ and $\Delta vrpB$, through gentamycin protection and competitive infection assays. We found that $\Delta creB\Delta vrpB$ was more attenuated than $\Delta vrpB$ (Supplementary Fig. 6k,l of the revised manuscript), indicating that the contribution of CreB to STM virulence is partially dependent on VrpB. Since CreB is a global regulator that controls the expression of a number genes, it is highly likely that CreB controls STM virulence through, in addition to VrpB, several other pathways, which need to be explored in future studies. This information is now added in the revised manuscript (lines 460–464).

Regarding the regulatory link between VrpB and SPI-2, we compared the virulence between $\Delta vrpB\Delta ssrB$ and $\Delta ssrB$. We found that the attenuated virulence of $\Delta vrpB\Delta ssrB$ was comparable to that of $\Delta ssrB$ (Supplementary Fig. 6m,n of the revised manuscript), indicating that the contribution of VrpB to STM virulence is solely dependent on SPI-2, which is consistent with the result (of the original manuscript) that complementation of $\Delta vrpB$ with SsrB restored bacterial virulence (Fig. 6j–l of the original manuscript). This information is now added in the revised manuscript (lines 464–468).

Please see below for more specific comments.

1. The title is Salmonella promotes its intracellular replication by reprogramming macrophage glucose metabolism..... which is very confusing. It speaks of reprogramming glucose metabolism to increase glycolysis and inhibit serine synthesis. The focus of the paper has to be rather on the identification of the virulence factors that sense metabolic changes.

Response:

Thank you for the comment. According to this comment and the No. 7 comment of reviewer #2, the title has been changed to “*Salmonella* Typhimurium senses macrophage metabolic changes mediated by T3SS effector SopE2 to promote intracellular replication and virulence”.

2. The authors have used Salmonella enterica serovar Typhimurium in their experiments which is one of the many serovars. Therefore, it is important to specify Salmonella Typhimurium in

the title. Salmonella alone is misleading.

Response:

Done.

3. It is also not clear to me why the authors address Salmonella as a model pathogen in the abstract when the whole manuscript is based on identifying virulence factors of Salmonella. In some places it is mentioned as Salmonella and in some as STM. It needs to be consistent.

Response:

We have deleted “The model pathogen” in the revised abstract (line 21). To improve consistently, we have now introduced STM at the beginning of the revised manuscript and used it throughout the revised manuscript.

4. What is the rationale for using peritoneal macrophages?

Response:

In this study, we need to use primary mouse macrophages to investigate the changes of intracellular metabolites upon *Salmonella* infection, because mouse macrophage cell lines (i.e. RAW264.7 or J774A.1) exhibit abnormal glucose metabolism (the Warburg effect) and are not suitable for metabolomics studies. The most commonly used mouse primary macrophages are peritoneal macrophages (PMs) and bone marrow-derived macrophages (BMDMs). PMs can be readily harvested from the mouse and have long been used to investigate the mechanisms of *Salmonella* intracellular survival and replication (eg. *Science*, 2000, 287:1655–8; *J Exp Med*, 2002, 195:1155–66; *J Exp Med*, 2005, 202(5):625–35) and also used for analysis of metabolic alterations upon LPS stimulation or bacterial infection (eg. *Cell Metab*, 2019, 29:1003–11; *Proc Natl Acad Sci U S A*, 2020, 117:15160–71). Moreover, at the beginning of this work, only the method for mouse PM collection was established in our laboratory and approved by the Institutional Animal Care Committee of Nankai University. Thus, we used PMs in this study.

BMDMs are more widely used than PMs in recent studies, because more cells can be obtained from a single mouse. The method for mouse BMDMs collection has now been established in our laboratory and approved by the Institutional Animal Care Committee of

Nankai University, we will use BMDMs in our future work.

5. In the infection protocol they mention that they pellet an overnight culture and use it to infect the macrophage monolayer. How do the authors know it is MOI 10? The infection protocol doesn't seem to be right. This is the basis for all the data.

Response:

Thank you for the comment. The macrophage infection assays were performed essentially as previously described (*Mol Microbiol*, 1998, 30:163–74; *Infect Immun*, 1989, 57:1–7).

The MOI 10 is defined as follows: Overnight-cultured bacterial strains (grown at 37°C in LB medium with shaking for 18 h) were pelleted by centrifugation at 5,000 rpm for 5 min, washed twice with PBS, re-suspended and adjusted to an OD₆₀₀ of 0.6 (0.60±0.01, ~5 × 10⁸ CFUs bacteria/mL) in RPMI medium. The bacterial culture was serially diluted to 4 × 10⁶ CFUs/mL and opsonized in RPMI medium containing 10% normal mouse serum for 20 min, and were then added to macrophage monolayers (0.5 mL/well) in the 24-well tissue culture plates (~2 × 10⁵ cells/well, cell number was quantified using LUNA-II™ automated cell counter).

We have now described the protocol in detail and added the relevant references in the Materials and Methods section of the revised manuscript (lines 642–649).

6. Moreover, the infection protocol for HeLa cells are very different from that of macrophages.

Response:

You are right that the infection protocol for HeLa cells differs from that of macrophages. The major difference is that the different growth states of STM used for infection of macrophages and HeLa cells (late stationary phase vs. log phase).

HeLa cells are non-phagocytic and inefficiently internalize *Salmonella*. Invasion of HeLa cells by *Salmonella* depends on the expression of SPI-1 T3SS, which is highly expressed during log phase while is not expressed during late stationary phase. Thus, the HeLa cells are generally infected with *Salmonella* which has been grown to log phase when the SPI-1 TTSS is highly expressed.

Macrophages actively engulf pathogens including *Salmonella*. The internalization of

Salmonella into macrophages does not require SPI-1 T3SS, and the expression of SPI-1 T3SS triggers rapid macrophage death (*Microbes Infect*, 2001, 3:1201–12). To prevent SPI-1-induced macrophage death, the macrophages are generally infected with *Salmonella* which has been grown to late stationary phase when the SPI-1 TTSS is not expressed.

To improve clarity, we have added the detailed information and the relevant references to the HeLa cell infection assays part in the Materials and Methods section of the revised manuscript (lines 665–668).

7. Salmonella Typhimurium are well known inducers of various cell death pathways. How do the authors correlate the metabolic data to cell death?

Response:

You are right that STM induces macrophage death during infection. It has been reported that STM can trigger both rapid and delayed macrophage death. *In vitro*, the rapid killing of macrophages by STM occurs within ~45 min of infection and depends on the expression of SPI-1 T3SS; the delayed killing of macrophages by STM requires prolonged infection (around 18–24 h post-infection) and depends on the expression of SPI-2 T3SS (*Cell Microbiol*, 2004, 6:1019–25).

In this work, we have designed our macrophage infection assays to avoid the effect of either kind of cell death on the metabolic results. To prevent SPI-1-induced rapid macrophage death, PMs were infected with STM grown to late stationary phase when the SPI-1 TTSS is not expressed. To prevent SPI-2-induced delayed macrophage death, we collected the metabolome samples at a time point (8 h post-infection) that well before the delayed macrophage death occurred (*Infect Immun*, 2001, 69:2293–301). In addition, we have carefully monitored the live cell number during the macrophage infection assays.

8. Peritoneal macrophages will be highly activated while isolating them due to the injection of sodium periodate. It is surprising that peritoneal macrophages withstood Salmonella infection for 20 h and the authors find glycolysis to be high.

Response:

You are right that the isolated PMs were activated due to the injection of sodium periodate. Using the protocol as described, a large proportion of the infected PMs (about 80%, according to the viable cell count at 0 and 20 hpi) withstood STM infection for 20 h. The same protocol used for PM isolation and infection was also utilized in many previously published reports (eg. *Science*, 2000, 287:1655–8; *J Exp Med*, 2002, 195:1155–66; *PLoS Negl Trop Dis*, 2015, 9:e3394).

9. Glucose uptake peaks at 8h pi but decreases by 20h – this observation has not been stated.

Response:

Our results showed that in STM-infected PMs, the glucose uptake rate at 20 hpi is similar to that of 8 hpi, and both are higher compared to that of 2 hpi (Extended Data Fig. 1a of the original manuscript). We have now clarified this in the revised manuscript (lines 123–125).

*10. How do the authors explain for the increase in succinate, fumarate and α -ketoglutarate although the transcriptomics data show that the respective enzymes such as *Sdha* are decreased? Increase in succinate also means that flux through TCA occurs if so, how does pyruvate accumulate?*

Response:

The metabolomics data showed that the levels of some TCA cycle intermediates, including succinate, fumarate, and α -Ketoglutaric acid (α -KG) were increased in STM-infected PMs compared to mock-infected PMs. This result is consistent with the previously published reports that succinate and fumarate were accumulated in LPS-stimulated BMDMs (eg. *Nature*, 2013, 496:238–42; *Cell Metab*, 2015, 21:65–80), and it was proposed that accumulation of some TCA cycle intermediates, especially succinate, is probably due to the decreased mitochondrial respiratory and reduced succinate dehydrogenase (SDH) activity in response to LPS stimulation or bacterial infection (*Nature*, 2013, 496:238–42; *Cell*, 2018, 174:780–4). In this work, the decreased expression of SDH (Fig. 1e) and the decreased mitochondrial respiratory rate (Fig. 1b) were observed in STM-infected PMs, which is in accordance with the metabolomics data. Thus, it is highly likely the increase in succinate was the result of reduced SDH activity while

the flux through TCA was decreased. We have now added an explanation for the increase in the TCA cycle intermediates in the revised manuscript to improve clarity (lines 156–158).

We speculate that the accumulation of pyruvate in STM-infected PMs might be due to the increase in glycolytic flux while decrease in TCA cycle activity, leading to increased pyruvate production but decreased pyruvate consumption.

11. The authors will have to present the original sea horse graph and data.

Response:

As requested, the original seahorse graph and data are now presented in the revised manuscript (Supplementary Fig. 1a,b of the revised manuscript).

12. For the next part of the data to show the effect of glycolysis authors have switched to RAW macrophages which is not the same as peritoneal macrophages and it is also not clear as why Pfkfb3 was silenced.

Response:

It is difficult to introduce siRNA into primary macrophages including PMs (*Clin Orthop Relat Res*, 1991, 181–91) and we have tried to introduce siRNA into PMs without success. Thus, in this study, we used RAW264.7 macrophages for the siRNA transfection experiments and used PMs for all the other mentioned work. The utilization of both primary macrophages and cell lines in one study is a common practice, and many published reports have used cell lines for siRNA transfection experiments while used primary macrophages for other experiments in the same study (eg. *Nature*, 2018, 563:714–8; *Cell Host Microbe*, 2017, 22:302–16).

Pfkfb3 is the key glycolytic rate-limiting enzyme that boosts glycolysis by producing fructose-2,6-bisphosphate and activating Phosphofructokinase 1. Many previously published reports blocked cellular glycolysis by silencing or inhibiting Pfkfb3 (eg. *FEBS Lett*, 2006, 580:3308–14; *J Exp Clin Cancer Res*, 2017, 36:7; *Gastroenterology*, 2020, 159:273–88).

In this study, to investigate whether the observed increase in macrophage glycolysis is associated with STM intracellular replication, we inhibited macrophage glycolysis *via* Pfkfb3 knockdown and showed that the knockdown significantly decreased STM replication in

RAW264.7 cells (Fig. 2a of the original manuscript). To improve clarity, we have added the sentence “To inhibit macrophage glycolytic activities,” prior to the original sentence “We used a small interfering RNA (siRNA) to decrease the expression of the key glycolytic enzyme Pfkfb3 in the murine RAW264.7 macrophage cell line” (line 171).

13. Bacterial burden is represented as fold change which is not appropriate. For instance, 0.2-fold change upon over-expressing Phgdh is not significant. Data needs to be represented as a log₁₀ CFU as has been shown for in vivo data.

Response:

We agree with the reviewer that the bacterial burden data would be better to be represented as a log₁₀ CFU. In this work, the bacterial burdens of mouse liver and spleen were represented as “log₁₀ CFU”, and bacterial burdens of *in vitro* intracellular replication in macrophages were represented as “fold intracellular replication”.

The reason to use fold change for STM intracellular replication in macrophages is mainly due to the fact that approximately 90% of the infected macrophages (*in vitro*) contains no more than ten bacteria. We believe that the “fold intracellular replication” can better reflect the differences in intracellular replication of STM in different macrophages *in vitro*. Moreover, “fold intracellular replication” is commonly used in recently published reports (eg. *Cell*, 2011, 144:675–88; *PLoS Pathog*, 2018, 14:e1007388; *Science*, 2018, 362:1156–60; *Nat Commun*, 2019, 10:3326; *mBio*, 2020,11:e03397–19).

Thus, we hope that we can still use fold change for STM intracellular replication in macrophages *in vitro*.

14. How does SopE2 transcriptionally repress phgdh?

Response:

SopE2 is a guanine nucleotide exchange factor whose primary target is the host Rho GTPase Cdc42 (*Mol Microbiol*, 2000, 36:1206–21). Through siRNA knock down of Cdc42, RT-qPCR analysis and gentamycin protection assays, we showed that the repression of *phgdh* by SopE2 was dependent on its action on Cdc42. However, the regulatory link between *phgdh* and Cdc42

is currently unclear. Cdc42 regulates diverse biological processes through its effects on the Jun amino-terminal kinase (JNK)/stress-activated protein kinase (SAPK) and p38 mitogen-activated protein kinase (MAPK) pathways. Whether Cdc42 regulates *phgdh* through one of these two known pathways or through an undiscovered pathway requires further study. We have added this information in the Discussion section of the revised manuscript (lines 537–538).

15. The authors claim PgtP as the transporter of G3P, pyruvate and lactate and they refer to an article from 1975. However, in the cited paper it is mentioned as PgtA. Is PgtP and PgtA synonymous?

Response:

Thank you for pointing out this error. Hong's group found that the *pgt* gene cluster encodes the transport system for PEP, 2PG, and 3PG in 1975 (*J Biol Chem*, 1975, 250:5089–96), and we referred to this paper in our original manuscript. The Reviewer is right that this paper did not work out functions of each individual gene in the *pgt* gene cluster.

In 1988, the same group found that gene *pgtP* within the *pgt* gene cluster is the transporter gene (*J Bacteriol*, 1988, 170:4304–8), and the other three genes *pgtA*, *pgtB*, *pgtC* in the *pgt* gene cluster are regulatory genes that necessary for *pgtP* gene expression (*J Bacteriol*, 1988, 170:3421–6). We have now referred to the same group's second paper (their year 1988 paper) in the revised manuscript (lines 976–982).

As mentioned in the original manuscript, the bacterial transport systems for pyruvate and lactate are CstA-YbdD and LldP, respectively.

16. Data shows that reduced glucose in RPMI increases the expression of PgtP. When authors performed metabolite analysis macrophages should have been grown in RPMI which contains 2g/L under such conditions PgtP expression is repressed. It is then contradictory that PgtP is the virulence factor that enables the bacteria to survive in macrophages by utilising G3P.

Response:

Sorry for the confusion.

i) The Reviewer is right that we performed metabolite analysis using macrophages which

have been grown in RPMI medium containing 2 g/L glucose. Our results showed that compared with the bacteria culturing in RPMI medium containing 2 g/L glucose, *pgtP* expression was highly increased when STM was in macrophages while the STM-infected macrophages were cultured in RPMI medium containing different levels of glucose from 0.25 to 2 g/L (Extended Data Fig. 3d in the original manuscript). This result indicates that STM upregulates its *pgtP* expression when STM is in macrophages, and this is consistent with that PgtP is a virulence factor involving in STM intracellular replication.

ii) Previous reports indicated that the available glucose is limited (at the micromolar level) inside the macrophages (cultured with DMEM containing 1 g/L glucose, ~5.5 mM) during STM infection (*PLoS Pathog*, 2013, 9:e1003301; *Cell Host Microbe*, 2013, 14:171–82). The induction of bacterial *pgtP* gene by lower level of glucose in macrophages is well consistent with those previous reports.

iii) Our results also showed that when PMs carrying STM were grown in RPMI containing different levels of glucose (2, 0.5, and 0.25 g/L), the expression of *pgtP* was induced earlier and at a higher level in PMs cultured with RPMI medium containing glucose of lower concentration (0.5 and 0.25 g/L) than in PMs cultured with RPMI medium containing glucose of higher concentration (2 g/L) (Extended Data Fig. 3c in the original manuscript). It is highly likely that lower glucose concentration in the medium resulted in its lower concentration within macrophages.

We have now modified the text to reflect this (lines 374–379).

*17. Is the macrophage metabolism affected when macrophages are infected with the *pgtP* mutant?*

Response:

Thank you for the comment. Although the replication ability of Δ *pgtP* in macrophages decreased (decreased by 2.8-fold compared to wt), it still displayed moderate intracellular growth, as the number of recovered CFUs of Δ *pgtP* in PMs increased 1.7-fold between 2 and 20 hpi (Fig. 3a of the original manuscript). We have now investigated the levels two key metabolites in Δ *pgtP*-infected PMs, lactate and 3PG. The levels of lactate and 3PG showed no

significant difference between $\Delta pgtP$ -infected and wt-infected PMs, and were significantly increased comparing to the mock-infected PMs (Supplementary Fig. 3d of the revised manuscript). As intracellular STM accounted for no more than 0.1% of the total metabolite concentrations, it is rational that the 3PG level in $\Delta pgtP$ -infected PMs is comparable to that of the wt-infected PMs, while $\Delta pgtP$ cannot utilize macrophage-derived 3PG. The results are now added in the revised manuscript (lines 236–240).

18. There is no substantial evidence that supports the claim that reprogramming glycolysis increases glycolysis.

Response:

Thank you for the comment. We have now performed lactate production assays to confirm that STM infection increases macrophage glycolysis. The results showed that STM-infected PMs exhibited a significantly increased lactate production at 2, 8, and 20 hpi, compared to mock-infected PMs (Supplementary Fig. 1c of the revised manuscript). We have now added this result in the revised manuscript (lines 119–120).

The combined results of lactate production assays, glucose uptake assays, ECAR and OCR assays, and also metabolomics and transcriptomics analyses collectively support the conclusion that macrophage glycolysis was increased upon STM infection.

19. Is the expression of $pgtP$ altered when $pfkfb3$ is silenced?

Response:

Through RT-qPCR analysis, we have now measured the expression levels of $pgtP$, $vrpA$, and crp in Pfkfb3 siRNA-treated and control siRNA treated RAW264.7 cells. The results showed that knockdown of Pfkfb3 resulted in significantly decreased $pgtP$, $vrpA$, and crp transcription of the intracellular STM (Supplementary Fig. 5h of the revised manuscript). The decreased STM $pgtP$, $vrpA$, and crp transcription indicates that bacterial glucose uptake was increased while bacterial 3PG uptake was decreased in Pfkfb3 siRNA treated RAW264.7 cells. Although the increase in glucose uptake is beneficial to STM replication in macrophages, the decreased 3PG utilization and SPI-2 gene expression (see below) by Pfkfb3 knockdown may have

collectively resulted in the reduced STM intracellular replication (Fig. 2a of the original manuscript).

The knockdown of Pfkfb3 resulted in significantly decreased SPI-2 gene expression, as shown by the decreased *creB*, *vrpB*, and *ssaG* transcription of the intracellular STM (Supplementary Fig. 6c of the revised manuscript). It is known that knockdown of Pfkfb3 decreases cellular glycolysis and lactate production (*FEBS Lett*, 2006, 580:3308–14; *J Exp Clin Cancer Res*, 2017, 36:7; *Gastroenterology*, 2020, 159:273–88). The decrease in STM *creB*, *vrpB*, and *ssaG* transcription is consistent with the decreased macrophage lactate production upon Pfkfb3 knockdown.

These results are now described in the revised manuscript (lines 383–386, 417–420, 477–479).

20. It would be convincing if the authors show the expression of the CreB 2 component system using reporter plasmids in macrophages.

Response:

Thank you for the suggestion. To determine whether the protein levels of CreB and CreC are increased inside macrophages, we constructed a *creBC* promoter-*lux* fusion (*creBC-lux*) and then compared its luminescence intensity inside PMs at 2, 8, and 20 hpi to that in RPMI medium. The results showed that the luminescence intensity of *creBC-lux* inside PMs was significantly increased at each time point post-infection (Supplementary Fig. 6a of the revised manuscript), confirming the increased expression of CreBC during STM growth in macrophages. The new results are now described in the revised manuscript (lines 397–402).

21. 10⁴ bacteria have been injected i.p for in vivo experiments. To my knowledge it is too high for the animals to survive even 6-7 days.

Response:

The reviewer is right. When i.p. infected with 1×10^4 CFUs of STM strain, most of the infected mice died between 3 and 7 days (eg. Fig. 3b of the original manuscript). The BALB/c mice (6–8 weeks old) used in this study were purchased from Beijing Vital River Laboratory

Animal Technology (Beijing, China). We have conducted substantial preliminary experiment to determine the optimal infection dose of STM, by evaluating survival and bacterial burdens in liver and spleen of mice infected with different doses of STM (10^3 , 10^4 and 10^5 CFUs), before choosing 1×10^4 CFUs as the infection dose for i.p infection in this work. Under this infection dose, the survival kinetics for mice infected with wt STM were similar to many other previously published reports (eg. *PLoS Pathog*, 2009, 5:e1000306; *Proc Natl Acad Sci U S A*, 2015, 112:5183-8; *mBio*, 2020, 11:e03397-19), and are differed significantly from that infected with mutant strains.

The bacterial dose of STM strain 14028s used for i.p infection of mice (6–8 weeks old) varied greatly, from 10^2 to 10^6 CFUs, among previously published reports when studying STM pathogenesis (eg. *Elife*, 2015, 4:e06792 (1×10^5 CFUs); *Nucleic Acids Res*, 2017, 45(17):9976–89 (1×10^6 CFUs); *PLoS Pathog*, 2018, 14:e1007388 (100–200 CFUs)).

22. Finally, Salmonella Typhimurium 14028s has been reported as a less invasive strain as they lack SPI-1 virulence factors such as SopE. How does this compare to a more invasive strain such as SL1344?

Response:

14028s and SL1344 are two commonly used mouse-virulent STM strains for studying *Salmonella* pathogenesis. Strain 14028s is less invasive than SL1344 due to the lack of SPI-1 virulence factors such as SopE. SopE is known to induce membrane ruffling of non-phagocytic cells through activating the Cdc42. However, SopE is absent from most STM strains, and the activation of Cdc42 in these STM strains are triggered by SopE2, which shows 69% sequence similarity and have a similar function to SopE (*Mol Microbiol*, 2000, 36:1206–21). Since SopE2 is common to all STM strains, it is highly likely that results obtained using 14028s in this study would be applicable to other STM strains including SL1344.

We have now performed additional experiments to confirm several key points of our findings in SL1344.

i) SL1344 infection increased macrophage 3PG, pyruvate and lactate levels. 3PG, pyruvate and lactate levels were measured for the PMs that infected by SL1344 for 8 h and the mock-infected PMs (NT).

ii) Cdc42 knockdown decreased SL1344 intracellular replication. Fold intracellular replication of SL1344 WT in Cdc42 siRNA-treated or control siRNA-treated RAW264.7 cells was expressed as the intracellular CFUs recovered at 20 hpi relative to those at 2 hpi.

iii) Utilization of 3PG is required for SL1344 replication in macrophages and virulence in mice. Fold intracellular replication of SL1344 WT and $\Delta pgtP$ in PMs was expressed as the intracellular CFUs recovered at 20 hpi relative to those at 2 hpi. Bacterial burden in the liver and spleen of SL1344 WT-infected mice and $\Delta pgtP$ -infected mice was determined at day 3 post infection.

iv) Lactate and pyruvate enhanced the transcription levels of SL1344 *creB* and SPI-2 genes. SL1344 *creB*, *vrpB*, and *ssaG* gene transcription in response to 1 mM pyruvate or lactate was determined by RT-qPCR analysis. SL1344 WT was grown in N-minimal medium in the presence or absence of 1 mM pyruvate or lactate for 6 h and then collected for RNA isolation and RT-qPCR analysis.

Reviewer #2 (Remarks to the Author):

In this study, Jiang, Wang, Song, et al investigate how the enteric pathogen Salmonella Typhimurium manipulates macrophage metabolism to support intracellular replication. Consistent with previous results, the authors provide evidence that in peritoneal macrophages, Salmonella induces an upregulation of flux through glycolysis, inhibition of serine synthesis, and decreased activity in the TCA cycle. Manipulation of macrophages with siRNA and inhibitors recapitulated these effects in the absence of Salmonella infection, and enhanced Salmonella replication. A mutant unable to transport 3-phosphoglycerate (pgtP) is attenuated for macrophage replication in vitro and in a mouse infection model. The authors identify the type three secretion system effector SopE2 as the cause for Salmonella-induced changes in serine biosynthesis. Furthermore, the authors uncover evidence suggesting that there are two novel Salmonella signaling pathways: low glucose levels result in CRP/cAMP-dependent transcriptional activation of vrpA, which in turn enhances transcription of pgtP. Lactate and pyruvate are sensed by the CreCB system, which promotes T3SS-2 transcription via VrpB. The authors conclude that inside macrophages, Salmonella inhibits serine synthesis to acquire the accumulated metabolite pre-cursors.

The topic of metabolism in the context of host-microbe interactions is timely and should be of great interest. There are several original findings and the concept that a type three secretion effectors alters host metabolism, which in turn facilitates pathogen growth, is intriguing. The writing is clear and the data presented in a logical manner. I have two major concerns regarding the validity of the overall conclusions:

1. The importance of SopE2 inhibiting serine synthesis in the mouse model is unclear. If the author's model is correct, one would predict that the SopE2 mutant is as attenuated as the other mutants tested in this study (e.g. PgtP). Since this is a central tenet of this work, the mutant is available, and the mouse model is established, this key experiment should be performed.

Response:

Thank you for your patient and thoughtful reading as well as the constructive comments and advices about our manuscript.

Thank you for the thoughtful suggestion. We have now measured the virulence of STM wt, $\Delta sopE2$, and $sopE2$ complemented strains in BALB/c mice through i.p. infection. Compared with mice infected with wt STM, mice infected with $\Delta sopE2$ exhibited a significantly increased survival rate and reduced bacterial burdens in the liver and spleen (Supplementary Fig. 4b,c), indicating that SopE2 contributes to STM systemic infection and the suppression of macrophage serine synthesis by SopE2 is necessary for STM systemic infection. The results are now presented in the revised manuscript (Supplementary Fig. 4b,c, lines 306–308)

However, as indicated by the results of gentamycin protection assays and *in vivo* competition assays, $\Delta sopE2$ is less attenuated than $\Delta pgtP$ (Supplementary Fig. 4d,e of the revised manuscript). SopE2 promotes STM virulence through increasing the accumulation of intracellular 3PG that can be used by STM as a carbon source. However, 3PG was also accumulated in $\Delta sopE2$ -infected PMs compared to that of the uninfected PMs (Fig. 4f of the original manuscript). The mutation of $pgtP$ entirely abrogated the utilization of 3PG by intracellular STM, while $\Delta sopE2$ can still use the macrophage-derived 3PG for intracellular growth, due to the function of PgtP. Thus, it is rational that $\Delta pgtP$ is more attenuated than $\Delta sopE2$. The results are now presented in the revised manuscript (Supplementary Fig. 4d,e, lines 309–318).

2. I have significant concerns regarding the interpretation of the experiments involving regulatory mutants, i.e. the $vrpA$ and $vrpB$ mutant, in the mouse model. The authors provide evidence that $VrpA$ regulates $PgtP$ transcription and $VrpB$ influences $SPI2$ transcription. However, it is very common that regulatory proteins affect more than one target. For example, the $vrpB$ mutant could be attenuated (Fig. 6k and l) because $SPI2$ or other (metabolic) genes are poorly expressed in general and not because $SopE2$ expression is affected. Similarly, the $vrpA$ mutant could be attenuated through action of another, unknown pathway. One simple approach to delineate cause-end-effect relationships and exclude these pleiotropic effects is through epistasis experiments. For example, one would expect that introducing a $vrpA$ mutation

into a *pgtP* mutant background (*vrpA pgtP* double mutant) should not further attenuate the strain. In contrast, if the *vrpA pgtP* double mutant was more attenuated than the *vrpA* mutant or the *pgtP* mutant, this would indicate off-target, pleiotropic effects or inconsistencies in the model. As such, the fold replication and the virulence in a mouse model for the following double mutants must be determined: *vrpA pgtP*, *creCB vrpB*, *vrpB SPI2*, and *PgtP SopE2*. In the absence of these data, several major conclusions are insufficiently supported.

Response:

Thank you for the comment. We have now generated the four double mutant strains: $\Delta vrpA \Delta pgtP$, $\Delta sopE2 \Delta pgtP$, $\Delta creB \Delta vrpB$, and $\Delta vrpB \Delta ssrB$. By carrying out gentamycin protection assays and competitive infection assays, we have now investigated the replication and virulence of $\Delta vrpA \Delta pgtP$ and $\Delta sopE2 \Delta pgtP$ compared to $\Delta pgtP$, $\Delta creB \Delta vrpB$ compared to $\Delta vrpB$, and $\Delta vrpB \Delta ssrB$ compared to $\Delta ssrB$.

We observed that the intracellular replication and virulence of $\Delta sopE2 \Delta pgtP$ and $\Delta vrpB \Delta ssrB$ was comparable to that of $\Delta pgtP$ and $\Delta ssrB$, respectively (Supplementary Fig. 4d,e and Supplementary Fig. 6m,n of the revised manuscript), indicating that the contributions of SopE2 and VrpB to STM virulence are entirely dependent on PgtP and SPI-2, respectively.

However, $\Delta vrpA \Delta pgtP$ was more attenuated than $\Delta pgtP$ (Supplementary Fig. 5a,b of the revised manuscript), indicating that the contribution of VrpA to STM virulence was partially dependent on PgtP, which is consistent with the result (of the original manuscript) that complementation of the *vrpA* with *pgtP* partially restored bacterial virulence (Fig. 5b,c,d of the original manuscript). Moreover, $\Delta creB \Delta vrpB$ was more attenuated than $\Delta vrpB$ (Supplementary Fig. 6k,l of the revised manuscript). These results indicate that VrpA and CreB also regulate other factors related to STM virulence.

These results are now presented and discussed in the revised manuscript (lines 309–318, 340–344, 460–468).

Specific Comments:

3. *Macrophage reprogramming and utilization of glucose by Salmonella has been reported before by Eisele et al. (PMID: 23954156). Their findings align very well with the findings in this report and therefore, it is unclear to me why the Eisele paper is not cited. It certainly should be.*

Response:

Sorry for having omitted this important reference in the original manuscript. We have now cited this paper in the revised manuscript (ref. 28 of the revised manuscript).

4. *Line 187: Should read “RT-qPCR” instead of “qPCR” and “transcription” instead of “expression”?*

Response:

“qPCR” has been changed to “RT-qPCR” throughout the revised manuscript. Related to the fold change in target gene at mRNA levels, “expression” has been changed to “transcription” throughout the revised manuscript.

5. *Line 240: The title of this section is a bit of an overstatement. This is only based on the analysis of two types of mutants and one time point. It is conceivable that other, unknown transporters of glucose exist and were not considered. Also, one has to consider the possibility that changing metabolism creates pleiotropic effects, such as the accumulation of toxic metabolites, which can affect fitness. My recommendation is to rephrase that section title.*

Response:

The title has been changed to “Macrophage-derived 3PG is an important carbon source for STM intracellular replication” in the revised manuscript (lines 251–252). The text in this section has been modified accordingly (line 274).

6. *Line 356 and Fig 7: Should read “cue” and not “signal”. It is unlikely that the mammalian host has evolved to produce pyruvate and lactate to communicate with bacteria. There are likely other instances in the manuscript where signal should be replaced with cue.*

Response:

Thank you for the suggestion. “signal” has been changed to “cue” throughout the revised manuscript.

7. The authors should consider rewording the title. Most of the manuscript is devoted to SopE2-mediated reprogramming of metabolism and coordinating bacterial gene expression.

Response:

Thank you for the comment. According to this comment and the No. 1 comment of Reviewer #1, the title has been changed to “*Salmonella* Typhimurium senses macrophage metabolic changes mediated by T3SS effector SopE2 to promote intracellular replication and virulence”.

Reviewer #3 (Remarks to the Author):

In this manuscript, Jiang et al. describe interesting findings showing that Salmonella Typhimurium (STM) modify glucose metabolism of macrophage to promote its intracellular replication and thus its virulence. Specifically, the authors demonstrate that STM upregulates the expression of enzymes for glycolysis and decreases the expression of enzymes for serine synthesis, which leads to the accumulation within macrophages of glycolytic intermediates, such as 3-phosphoglycerate (3PG), pyruvate and lactate. The mechanism by which STM represses serine synthesis, as well as a mechanism for the uptake of macrophage 3PG to be used by STM as a carbon source, which involves sensing of low glucose, were defined. The authors also demonstrate that STM senses pyruvate and lactate to induce expression of the SPI-2 genes, which are known to be essential for STM intracellular replication, and that the regulatory cascade responding to pyruvate and lactate is required for STM virulence.

I find this study as technically sound and relevant, with novel results, as well as the manuscript well written. The findings from this study further expand the knowledge about the different mechanisms used by pathogenic bacteria to manipulate hosts; in this case, by reprogramming metabolism.

However, I recommend to consider findings from some other reports for the integration of a more complete picture about the mechanisms used by Salmonella to survive/replicate inside macrophages. Additionally, the role of the pyruvate/lactate-CreB-VrpB-SPI2 regulatory cascade in Salmonella intracellular replication should be further assessed. See comments below.

Comments

1. Whereas this study shows that STM infection enhances glycolysis in macrophages, the report by Sanman et al. (2016) indicates that STM infection reduces glycolytic flux also in

macrophages. Please discuss about this discrepancy.

Response:

Thank you for your patient and thoughtful reading as well as the constructive comments and advices about our manuscript.

In our work, the combined results of ECAR and OCR assays, lactate production assays, glucose uptake assays, and metabolomics and transcriptomics analyses collectively support the conclusion that macrophage glycolysis is increased upon STM infection. Sanman *et al.* (2016) showed that the lactate production rate in BMDMs was reduced upon STM infection from 5 to 11 hpi. However, the overall lactate levels in STM-infected BMDMs at both 5 and 11 hpi were higher than that of the uninfected cells, which is well consistent with our results, indicating macrophage glycolysis is elevated upon STM infection.

2. How is increased the expression of PMs glycolytic genes? Is SopE also involved in their upregulation?

Response:

Complex signaling pathways are involved in the upregulation of macrophage glycolytic genes upon bacterial infection, with the stabilization and accumulation of HIF-1 α plays a critical role. Under normal conditions, HIF-1 α is rapidly degraded because of hydroxylation of HIF-1 α by prolyl hydroxylase (*J Bioenerg Biomembr*, 2007, 39:223–9). Following recognition of bacterial PAMP by macrophage Toll-like receptors (TLRs) (eg, TLR4 agonist LPS), the activation of the PI3K/ AKT/mTOR pathway by TLR4 lead to HIF-1 α stabilization. Upon stabilization of HIF-1 α , it binds to HIF-1 β to form HIF-1 complex. The HIF-1 complex transactivates many key glycolytic genes, including *slc2a1*, *pfkfb3*, and *ldha* (*Cell Mol Life Sci*, 2018, 75:2093–109). Moreover, the glycolytic end products pyruvate and lactate induce HIF-1 α accumulation, which in turn increases HIF-1 activity to increase the expression of glycolytic genes (*J Biol Chem*, 2002, 277:23111–5).

Through RT-qPCR analysis, we have now tested whether SopE2 related with the regulation of two key macrophage glycolytic genes. The transcription levels of *pfkfb3* and *slc2a1* showed

no significant difference between ΔsopE2 -infected and wt-infected PMs, and were significantly increased comparing to the mock-infected PMs (Supplementary Fig. 4a of the revised manuscript), revealing SopE2 does not have a regulatory role on macrophage glycolytic gene expression. These results are now described in the revised manuscript (lines 286–287).

3. Are there previous studies involving Cdc42 on gene expression or specifically on regulation of genes for metabolism, such as those for serine synthesis?

Response:

Our results indicate that STM represses the macrophage serine synthesis pathway through a SopE2-mediated Cdc42 mechanism. Cdc42 is a small Rho GTPase regulating multiple biological processes in eukaryotic cells, including actin cytoskeletal organization, gene transcription, cell proliferation and differentiation (*Nature*, 2002, 420:629–35). Cdc42 regulates biological processes through its effects on the Jun amino-terminal kinase (JNK)/stress-activated protein kinase (SAPK) and p38 mitogen-activated protein kinase (MAPK) pathways (*Cell*, 1995, 81:1137–46; *J Biol Chem*, 2000, 275:26441–8). However, there was no previous studies indicating that Cdc42 involved in the regulation of genes for metabolism and those for serine synthesis.

4. It is unclear how VrpA was identified as a regulator of pgtP by using BPRM, a tool for prediction of bacterial promoters. Same for CRP as a regulator of vrpA. Please describe with more detail this procedure.

Response:

Thank you for pointing it out. BPRM is a bioinformatic tool for bacterial promoter prediction that has been used in many promoter-related studies. In addition to predict promoter, this program can also provide information about the binding sites of known transcription factor(s) in the predicted promoter region. Using BPRM, a putative CRP-binding site was identified in the *vrpA* promoter region, indicating CRP might be a potential regulator of VrpA. We have modified the text to improve clarity (lines 350–352).

It was a mistake to describe that “The putative upstream regulators of *pgtP* were predicted

using the program BPROM” in our original manuscript, and thank you very much for pointing out this mistake. The regulation of *pgtP* by VrpA was not identified by BPROM. Upon investigating the regulatory functions of 18 putative regulatory proteins that present only in STM but not in *Escherichia coli* (Supplementary Table 3 of the revised manuscript), we identified VrpA as a potential positive regulator of *pgtP*, as the transcription of *pgtP* and the other three genes in the *pgt* gene cluster were downregulated in the *vrpA* mutant. We have now corrected this mistake in the revised manuscript (lines 324–330) and presented the transcriptome of *vrpA* mutant data in the revised manuscript (and deposited in the NCBI Sequence Read Archive, PRJNA561041, <https://www.ncbi.nlm.nih.gov/sra/PRJNA561041>).

5. What kind of regulator is VrpA? Is it controlled by any of the regulators involved in the expression of genes for STM intracellular replication? You could easily check this in SalComRegulon.

Response:

Thank you for the comment. VrpA encodes a XRE family transcriptional regulator (118 amino acid length) with a predicted DNA-binding helix-turn-helix motif as predicted by BLAST.

We took this advice and analyzed expression profiles of VrpA using SalComRegulon. It was found that *vrpA* transcription is down regulated 2.3-fold in a ferric uptake regulator (Fur) mutant in comparison to wt STM, indicating that *vrpA* may also be positively regulated by Fur. As CRP is a much stronger positive regulator (16.4-fold), it is highly likely that CRP acts dominantly on the transcription of *vrpA*.

The exact function of Fur for STM replication inside macrophages is unclear at present. However, Fur is reported to be required for STM systemic infection of mice (*J Bacteriol*, 2011, 193:497–505) and the expression of *fur* is upregulated when STM is inside macrophages (*PLoS Pathog*, 2015, 11(11):e1005262; *Science*, 2018, 362:1156–1160). Many of the Fur-regulated systems including iron acquisition, acid adaptation, oxidative stress resistance, are related with the adaptation of STM to macrophage environment. Future studies are required to illustrate the regulatory function of Fur inside macrophages and its relationship with VrpA.

6. *Is CRP required for intracellular replication of STM? Discuss about what is known on the role of CRP in Salmonella virulence.*

Response:

Thank you for the comment. In the original manuscript, through i.p.-infected mice, we showed that mice infected with Δcrp exhibited a markedly reduced bacterial burdens in the liver and spleen, revealing CRP might influence STM intracellular replication. Through gentamycin protection assays, we have now investigated the influence of *crp* mutation on STM intracellular replication. The results showed that *crp* mutation significantly decreased STM replication in PMs (Supplementary Fig. 5c of the revised manuscript), indicating CRP is required for STM intracellular replication. The results are now described in the revised manuscript (lines 353–355).

As requested, we discussed about the role of CRP in STM virulence based on the findings from this study and previous reports in the revised manuscript (lines 570–576). “*The cAMP-CRP of STM in response to macrophage glucose limitation activates the expression of PgtP via VrpA. The increased expression of PgtP enables intracellular STM to utilize macrophage-derived 3PG as a major carbon source. It is known that cAMP-CRP also induces the transcription of SPI-2 genes⁵⁹. Thus, it is possible that cAMP-CRP integrates the utilization of macrophage-derived 3PG with the enhancement of SPI-2 gene expression to achieve optimal intracellular replication of STM.*”

7. *Consider for Discussion previous findings reported by Westermann et al. (2016; doi: 10.1038/nature16547), showing the effect of STM infection on macrophage transcriptome, including data indicating repression of SopE2, SPI-2 and CRP by PhoP-PinT when Salmonella is intracellularly. How these findings and the results from this study can converge to explain all the mechanisms used by STM for intracellular replication?*

Response:

Thank you for the comment. We have now cited this paper and discussed this work in the revised manuscript (lines 576–580). In addition, we have discussed and related our findings

with previous finds from other studies, and described a possible mechanism used by STM for intracellular replication in the revised manuscript (lines 566–580).

“When STM replicates within macrophages, macrophage glycolytic pathway is increased and STM uses the T3SS effector SopE2 to repress macrophage serine synthesis pathway, resulting in the accumulation of glycolytic intermediates in macrophages, including 3PG, pyruvate, and lactate. The two-component system CreBC of STM senses the increased pyruvate and lactate levels of macrophages to activate the expression of SPI-2 genes via VrpB. The cAMP-CRP of STM in response to macrophage glucose limitation activates the expression of PgtP via VrpA. The increased expression of PgtP enables intracellular STM to utilize macrophage-derived 3PG as a major carbon source. It is known that cAMP-CRP also induces the transcription of SPI-2 genes⁵⁹. Thus, it is possible that cAMP-CRP integrates the utilization of macrophage-derived 3PG with the enhancement of SPI-2 gene expression to achieve optimal intracellular replication of STM. However, it is worth noticing that a PhoP-activated small RNA PinT has been shown to repress sopE2, crp, as well as SPI-2 genes when STM is inside macrophages⁶³, suggesting the presence of negative regulatory mechanism to repress the overexpression of these virulence factors. How those positive and negative regulatory systems act in conjunction to achieve optimal STM intracellular replication is unclear and requires future studies.”

8. While it is clear that the regulatory pathway CreB-VrpB induces expression of the SPI-2 genes in response to pyruvate or lactate, and that it is required for STM intracellular replication, the STM mutants blocked in the uptake of pyruvate or lactate are not affected in intracellular replication, which could be due to the fact that pyruvate and lactate probably activate by separate the CreB-VrpB regulatory cascade. To further investigate the role of the pyruvate/lactate-CreB-VrpB-SPI2 regulatory cascade in Salmonella virulence, you can test a STM double mutant, blocked in the uptake of both pyruvate and lactate, for intracellular replication and for activation of SPI-2 (ssaG) expression in response to pyruvate and lactate. This could make stronger your conclusion indicating that the SPI-2 genes are induced inside macrophages in response to pyruvate or lactate for intracellular replication and thus for

virulence of Salmonella. Is there any domain in CreC that could be involved in sensing pyruvate and/or lactate?

Response:

Thank you for the comment. We have now generated the *lldp cstA-ybdD* double mutant, which cannot uptake either pyruvate or lactate, and compared the intracellular replication and virulence between $\Delta lldp \Delta cstA-ybdD$ and wt. We also tested the *ssaG* expression in $\Delta lldp \Delta cstA-ybdD$ in response to pyruvate and lactate. The results showed that the replication ability of $\Delta lldp \Delta cstA-ybdD$ in PMs and the virulence of $\Delta lldp \Delta cstA-ybdD$ in mice were comparable to that of the wt (Supplementary Fig. 3e–g of the revised manuscript), revealing that the utilization of host-derived pyruvate and lactate is not essential for STM intracellular replication and virulence. Through RT-qPCR analysis, we found that *lldp* and *cstA-ybdD* double mutation did not influence *ssaG* transcription in the presence of pyruvate and lactate when STM grown in N-minimal medium (Supplementary Fig. 6g of the revised manuscript), revealing that the utilization of pyruvate and lactate is not related with SPI-2 gene expression. These data further indicate that the SPI-2 genes are induced inside macrophages in response to pyruvate or lactate for intracellular replication and thus for virulence of STM. These new results are now described in the revised manuscript (lines 244–247, 430–433).

There is no previous report to study the domain of the CreC protein which responds to pyruvate and/or lactate. Pfam analysis (<http://pfam.xfam.org/>) indicated that CreC protein contains a Histidine Kinase A (dimerization/phosphoacceptor) domain (HisKA) and an ATP-binding domain (GHKL) in N-terminal. It is probably that these two domains are involved in the sensing pyruvate and/or lactate by CreC, which can be explored in future studies.

9. Integrate in your model the findings from this work and those from other studies. For instance, this work show that low glucose as well as pyruvate and lactate are cues detected by Salmonella inside macrophages through mechanisms involving the regulators CRP, VrpA, CreC/CreB and VrpB, being the CreC/B-VrP pathway necessary for induction of the SPI-2 genes; SPI-2 is essential for Salmonella replication within host cells. However, several other studies have demonstrated that Salmonella also senses other signals present inside macrophages, such as

acidic pH and low phosphate and magnesium, through regulators well characterized like PhoP/Q, OmpR/EnvZ and SsrA/B, which induce expression of the SPI-2 genes inside host cells.

Response:

Thank you for the comment. Fig. 7 presents a schematic of our findings. And we have now discussed those related findings in the revised manuscript (lines 545–550, 566–580).

Other changes:

1. The method used to measure lactate and pyruvate concentrations has now been added to the Materials and Methods section of the revised manuscript (lines 744–755).
2. The method used to generate the *creBC-lux* reporter fusion and the method used to determine intracellular *creBC* expression by using this fusion have now been added to the Materials and Methods section of the revised manuscript (lines 616–618, 807–815).
3. The “Abstract” section has been revised to meet the word limit requirement (lines 21–34).
4. The “Data Availability” section has been revised according to the requirement of *Nature Communications* (lines 866–870).
5. Full gels and blots of the western blots and EMSAs are provided in Supplementary Fig. 7 and Supplementary Fig. 8, respectively.
6. All bar graphs in the original manuscript have been replaced with plots that feature information about the distribution of the underlying data.

REVIEWERS' COMMENTS

Reviewer #1 (Remarks to the Author):

The authors have addressed most of my comments adequately.

1. Despite the glucose uptake being similar at 8h and 20h and lactate being increased, ECAR is reduced at 20h to the equivalent of 2h (Fig 1a). Can the authors explain?
2. Authors have reasoned for the increase in succinate. Not only succinate but also fumarate is increased despite SDH (enzyme that converts succinate to fumarate is decreased).
3. It will be good if the cell death data is provided.

Reviewer #2 (Remarks to the Author):

The authors have adequately addressed my concerns.

Reviewer #3 (Remarks to the Author):

All my concerns were resolved by authors. I have no further comments about this great study.
Víctor H. Bustamante

Responses to reviewers

Reviewer #1:

The authors have addressed most of my comments adequately.

Response:

Thank you for your consideration of this work. We appreciate your effort in helping us improve our manuscript. We have carefully revised the manuscript based on your comments and suggestions. Detailed point-by-point responses are provided below.

1. Despite the glucose uptake being similar at 8h and 20h and lactate being increased, ECAR is reduced at 20h to the equivalent of 2h (Fig 1a). Can the authors explain?

Response:

Thank you for the comment. Lactate production value (Supplementary Fig. 1c) indicates the levels of accumulated lactate in the cell supernatants of STM-infected PMs or mock-infected PMs at 2, 8, and 20 hpi. 2-NBDG (glucose) uptake value (Supplementary Fig. 1d) reflects the glucose uptake rate of STM-infected PMs or mock-infected PMs at the indicated time points (2, 8, or 20 hpi). ECAR value (Fig. 1a) reflects the lactate production rate of STM-infected PMs or mock-infected PMs at the indicated time points (2, 8, or 20 hpi).

The total amount of lactate accumulated in STM-infected PMs at 2, 8, and 20 hpi is 2.58, 7.51, and 9.77 $\mu\text{M}/(10^5 \text{ cells})$, respectively. By calculating, the amount of lactate increased 4.93 $\mu\text{M}/(10^5 \text{ cells})$ in the first 6 h (from 2 to 8 h), and only 2.26 $\mu\text{M}/(10^5 \text{ cells})$ in the later 12 h (from 8 to 20 h), indicating that the rate of lactate production in STM-infected PMs is decreased after 8 hpi compared to that before 8 hpi. This explains the decreased ECAR value, which reflects the rate of lactate production, at 20 h compared to that of 8 h (Fig. 1a).

2. Authors have reasoned for the increase in succinate. Not only succinate but also fumarate is increased despite SDH (enzyme that converts succinate to fumarate is decreased).

Response:

Thank you for the comment. When macrophages shift to a Warburg metabolism, the TCA cycle activity is decreased while glutamine metabolism is increased, and cells utilize glutamine as an anaplerotic carbon source to replenish TCA cycle intermediates (*Nature*, 2013, 496:238–42; *Cell Metab*, 2008, 7:11–20). In this process, glutamine is converted to α -ketoglutaric acid (α -KG), which enters the TCA cycle, leading to the increased levels of intracellular α -KG, succinate, and fumarate (*Nature*, 2013, 496:238–42).

The increases in the levels of α -KG, succinate and fumarate in macrophages observed in this study could be also attributed to glutamine-dependent anaplerosis. We have added the relevant explanation in the revised manuscript (lines 151–157).

3. It will be good if the cell death data is provided.

Response:

The cell death data is now provided in the revised Supplementary Fig. 1e and also described in the revised manuscript (lines 123–124).

Reviewer #2:

The authors have adequately addressed my concerns.

Response:

Thank you for your consideration of this work. We appreciate your effort in helping us improve our manuscript.

Reviewer #3:

All my concerns were resolved by authors. I have no further comments about this great study.

Victor H. Bustamante

Response:

Thank you for your consideration of this work. We appreciate your effort in helping us improve our manuscript.